# Epidemiological model for the inhomogeneous spatial spreading of COVID-19 and other diseases

Yoav Tsori[1,3]*, Rony Granek[2,3]

1 Department of Chemical Engineering, Ben-Gurion University of the Negev, Beer Sheva, Israel, 2 The Avram and Stella Goldstein-Gorren Department of Biotechnology Engineering, Ben-Gurion University of the Negev, Beer Sheva, Israel, 3 The Ilse Katz Institute for Nanoscale Science and Technology, Ben-Gurion University of the Negev, Beer-Sheva, Israel

* tsori@bgu.ac.il

**Data Availability Statement:** All figures are available from the medRxiv database (https://doi.org/10.1101/2020.07.08.20148767).

**Funding:** YT acknowledges support by the Israel Science Foundation Grant No. 274/19 (www.isf.

## Abstract

We suggest a novel mathematical framework for the in-homogeneous spatial spreading of an infectious disease in human population, with particular attention to COVID-19. Common epidemiological models, e.g., the well-known susceptible-exposed-infectious-recovered (SEIR) model, implicitly assume uniform (random) encounters between the infectious and susceptible sub-populations, resulting in homogeneous spatial distributions. However, in human population, especially under different levels of mobility restrictions, this assumption is likely to fail. Splitting the geographic region under study into areal nodes, and assuming infection kinetics within nodes and between nearest-neighbor nodes, we arrive into a continuous, "reaction-diffusion", spatial model. To account for COVID-19, the model includes five different sub-populations, in which the infectious sub-population is split into pre-symptomatic and symptomatic. Our model accounts for the spreading evolution of infectious population domains from initial epicenters, leading to different regimes of sub-exponential (e.g., power-law) growth. Importantly, we also account for the variable geographic density of the population, that can strongly enhance or suppress infection spreading. For instance, we show how weakly infected regions surrounding a densely populated area can cause rapid migration of the infection towards the populated area. Predicted infection "heat-maps" show remarkable similarity to publicly available heat-maps, e.g., from South Carolina. We further demonstrate how localized lockdown/quarantine conditions can slow down the spreading of disease from epicenters. Application of our model in different countries can provide a useful predictive tool for the authorities, in particular, for planning strong lockdown measures in localized areas—such as those underway in a few countries.

## Introduction

The COVID-19 pandemic is now spread over most of the globe. Its vast consequences are associated with severe public health issues, i.e. overwhelmed health system, high death toll, and a

org.il). The funders had no role in study design, data collection and analysis, decision to publish, or preparation of the manuscript.

**Competing interests:** The authors have declared that no competing interests exist.

huge economic crisis worldwide [1–5]. In order to optimize decisions in both aspects, health and economy, authorities need information and predictions about the spatial distribution of the disease [6], thereby allowing selective quarantine or lockdown measures [7, 8].

Infectious disease spreading models are largely based on the assumption of perfect and continuous "mixing", similar to the one used to describe the kinetics of spatially-uniform chemical reactions. In particular, the well-known susceptible-exposed-infectious-recovered (SEIR) model, builds on this homogeneous-mixing assumption. Some extensions of SEIR-like models that account for spatial variability employed mainly diffusion processes for the different sub-populations [9–12] (where the term sub-population refers to people under a certain stage of the disease). Yet, clearly, while such processes can effectively describe wildlife motion in some systems, they fail to describe the (non-random) human behavior [13]. To mimic human behavior more realistically, recent extensions employed diffusion processes of the sub-populations that are limited to contact networks [14]. However, one of the most important artifacts for the application of diffusion process for human population is its unrealistic tendency to spread all populations to uniformity (be it in real space or on contact networks). Moreover, these models do not involve naturally a spatial dependence of infection spreading parameters, which are required to model geographically local quarantine. Thus, to implement such dependence using homogeneous models requires a division of the geographic region into multiple number of patches [14, 15].

Early in the COVID-19 pandemic spread, different modeling groups used homogeneous models to predict the epidemic evolution in China and in other countries [16–23]; their predictions urged the World Health Organization (WHO) to issue a global warning. Wu *et al.* were the first to model the COVID-19 spreading [16]. They applied the SEIR model using data from the very early (exponential) stage of the outbreak, to predict epidemic spread mainly in Wuhan and mainland China. Extensions for this first attempt were quick to follow. Ivorra and Ramos applied the "Be-CoDiS" mathematical model—a multi-sub-population extension of the SEIR model—to COVID-19 [19, 20]. A fit of the parameters of the model to a longer period of evolution, up to the time where the outbreak nearly peaked (maximum number of new daily infected people), yielded remarkably accurate predictions for the stages that followed. More recently, He *et al.* [21] and Giordano *et al.* [22] provided further improvements and analysis on the original application of the SEIR model to COVID-19 [16].

As mentioned, conventional epidemiological models assume spatially uniform (statistical) frequency of encounters between infectious and susceptible people, which is associated with uniform spatial densities of these sub-populations at all times [16, 24, 25]. As such, these models do not require any spatial variable. However, the assumption of "infinitely fast mixing" might fail even in normal life conditions, let alone under (the often used) various travel and gathering restrictions, or moderate quarantine conditions [7]. Moreover, the basic reproductive number, $R_0$, may vary from one area to the other, for example, if people behave differently across areas. A further complication may arise when human behavior evolves in time during an epidemic, implying that $R_0$ is also time-dependent. As a consequence, accurate predictions of such models rely on repeated readjustments of the infection rate constant as the epidemic progresses.

Specifically, these models give a broad regime of exponential growth of the cumulative number of infected people, whereas the actual growth is sub-exponential, which can be effectively described by a power-law (i.e. $At^\nu$ where $A$ and $\nu$ are numerical constants) [26–30]. Early data from several countries indeed demonstrated such a wide temporal regime of power-law growth, which occurred much prior to the peak of the epidemic [26, 30–32]. In accord with similar ideas [26], we propose that such a temporal behavior can result from the lateral

spreading of infected domains, similar to the spreading of wildfire, which gives further motivation to the present work.

In this paper, we develop a novel theoretical framework to model the spread of infections, thereby allowing researchers working in close contact with authorities to apply the model in their own countries. To demonstrate our theoretical framework, we chose to improve on homogeneous SEIR-like models, in particular in the context of the COVID-19 pandemic, in three major aspects. The first is derivation of a spatial spreading (diffusion-like) operator, that generates propagation of the front of an infected population domain into a susceptible one [33–35]. Importantly, this operator is associated only with infection spreading and does not describe motion of the different populations as previously suggested [9, 12]; as argued above, the use of a diffusion process is inadequate for human population. The second, and strongly linked to the first, is the ability to account for the geographical population density variation and study its effect on the spreading [36]. The combination of these two aspects leads to the spreading of the disease from areas of low density to areas of high density. The third aspect is the account of geographic variation in quarantine levels if such are employed. In addition, to apply our approach for COVID-19, we split the infectious sub-population into a transient "infectious-presymptomatic" group and an "infectious-symptomatic" group (the SEPIR model mentioned below).

Our results provide infection "heat-maps", reminiscent of those appearing in publicly available resources (e.g., for South Carolina [37] and Nashville, Tennessee [38]); see also the heat-map snapshots in the SI, Figs SI-6 and SI-7 in S1 File. These heat-maps demonstrate unique features of the disease spreading depending on the spatial variation of population density, location of the initial epicenters, and local quarantine levels. They show that the accumulated number of infected people deviates significantly from the associated homogeneous model.

## *In*-homogeneous SEPIR model

Our model, which builds on the homogeneous SEIR model, includes five sub-populations associated with different stages of the disease: susceptible-exposed-presymptomatic-infectious-recovered (SEPIR). This basic SEPIR model is extended to account for the spatially varying density of people $n(\mathbf{x})$—where $\mathbf{x}$ is a 2-dimensional (2D) vector in the plane, whose components will be denoted by $x$ and $y$—between different geographical areas of the region under study, which is assumed in the present study to be isolated from other regions. Our model also aims to predict the effect of different quarantine levels imposed in different areas within the region of study, which is modeled via *spatial dependence* of infection rates.

We define the variables $h$, $b$, $w$, $f$, and $r$ as follows:

$h(\mathbf{x}, t)$: 2D (areal) density of *susceptible* (healthy but not immune) people

$b(\mathbf{x}, t)$: density of *exposed* people that do *not* yet infect others and are *not* yet symptomatic (i.e. within the incubation period [39])

$w(\mathbf{x}, t)$ density of *pre-symptomatic* people that can already infect others but are not yet symptomatic (i.e. still within the incubation period)

$f(\mathbf{x}, t)$: density of *symptomatic* (infectious) people.

$r(\mathbf{x}, t)$: density of people that have *recovered* from illness, thus assumed (here) to be immune from a second infection

We require that the total *local* density of people is equal to prescribed values $n(\mathbf{x})$ at different positions $\mathbf{x}$ (obtained, e.g., from public databases) and is *independent of time*:

$$h(\mathbf{x}, t) + b(\mathbf{x}, t) + w(\mathbf{x}, t) + f(\mathbf{x}, t) + r(\mathbf{x}, t) = n(\mathbf{x}) , \qquad (1)$$

implying the same for the total population number, $N = \int_{\text{area}} d^2 x\, n(\mathbf{x})$. In what follows, capital letters will denote the corresponding global quantities, e.g., $H(t) = \int_{\text{area}} d^2 x\, h(\mathbf{x}, t)$, $B(t) = \int_{\text{area}} d^2 x\, b(\mathbf{x}, t)$, etc. We note in passing that these variables correspond to the variables of the well-known (homogeneous) SEIR model [24, 40, 41] as follows: $H \leftrightarrow S$ (susceptible), $B \leftrightarrow E$ (exposed), $W + F \leftrightarrow I$ (infectious), $R \leftrightarrow R$ (recovered). Thus, unlike in the SEIR model, in our model the infectious sub-population ($I$) is split into two sub-populations, pre-symptomatic ($W$) and symptomatic ($F$).

In order to develop the spatial epidemic spread model, we consider first a 2D discrete space (square or triangular lattice), in which the nodes are defined as areal units of linear size (henceforth "grid-size") $\delta$. The present model assumes some traveling (or mobility) restrictions, and the value of $\delta$ is chosen such that node-node infections can occur only between those nodes that are nearest-neighbors, while within each node homogeneous infections take place. For example, if people avoid traveling distances over 30 km, but still travel a lot within 30 km, one has to choose $\delta \simeq 30$ km. Likewise, under stronger restrictions (i.e. a "lockdown"), traveling may be restricted to 1 km, which implies $\delta \simeq 1$ km.

We define by $h_i(t)$ the number of susceptible people at node $i$ at time $t$. Similarly $b_i(t)$, $w_i(t)$, $f_i(t)$, and $r_i(t)$ describe the numbers of the different sub-populations at each node. Infection can occur at a rate constant $k_1(i)$ when infectious and susceptible people from the same node $i$ meet each other, and at a rate constant $k_2(j, i) = k_2(i, j)$ when meetings occur between an infectious person from node $i$ and a susceptible person from a nearest-neighbor (NN) node $j$ (i.e. the infection rate, for an infectious person at node $i$ and a susceptible one at node $j$, is identical to the rate when the two nodes are interchanged). The total number of people at node $i$ is denoted by $n_i$. Accordingly, the set of (non-linear) master equations for the distribution of these sub-populations is

$$
\begin{aligned}
\frac{\partial h_i}{\partial t} &= -k_1(i)\frac{h_i}{n_i}(w_i + f_i) - \frac{h_i}{n_i}\sum_{j \in i} k_2(j, i)(w_j + f_j) \\[4pt]
\frac{\partial b_i}{\partial t} &= k_1(i)\frac{h_i}{n_i}(w_i + f_i) + \frac{h_i}{n_i}\sum_{j \in i} k_2(j, i)(w_j + f_j) - \gamma_0 b_i \\[4pt]
\frac{\partial w_i}{\partial t} &= \gamma_0 b_i - \gamma_1 w_i \\[4pt]
\frac{\partial f_i}{\partial t} &= \gamma_1 w_i - \gamma_2 f_i \\[4pt]
\frac{\partial r_i}{\partial t} &= \gamma_2 f_i \ .
\end{aligned}
\qquad (2)
$$

where $j \in i$ stands for node $j$ that is NN to $i$. In Eq (2), $\gamma_0$, $\gamma_1$, and $\gamma_2$ are the rate coefficients for the transition of the $b$-population to $w$, of $w$ to $f$, and of $f$ to $r$, respectively.

We now transform the master Eq (2) to the continuum using the Kramers-Moyal expansion [42], $\mathbf{x} \leftrightarrow i$. Using the symmetry for infection rates between NN nodes $i$ and $j$, $k_2(j, i) = k_2(i, j)$, and defining a local density of a sub-population $y$ as $y(\mathbf{x}, t) \equiv y_i(t)/\delta^2$ (e.g., $b(\mathbf{x}, t) \equiv b_i(t)/\delta^2$),

we obtain

$$
\begin{aligned}
\frac{\partial h}{\partial t} &= -k(\mathbf{x})\frac{h}{n}(w+f) - \frac{h}{n}\vec{\nabla}\left[D_k(\mathbf{x})\vec{\nabla}(w+f)\right] \\
\frac{\partial b}{\partial t} &= k(\mathbf{x})\frac{h}{n}(w+f) + \frac{h}{n}\vec{\nabla}\left[D_k(\mathbf{x})\vec{\nabla}(w+f)\right] - \gamma_0 b \\
\frac{\partial w}{\partial t} &= \gamma_0 b - \gamma_1 w \\
\frac{\partial f}{\partial t} &= \gamma_1 w - \gamma_2 f \\
\frac{\partial r}{\partial t} &= \gamma_2 f \ .
\end{aligned}
\tag{3}
$$

where the last line in Eq (3) can be replaced—using the conservation law Eq (1)—by $r = n - (h + b + w + f)$. In Eq (3), $k(\mathbf{x}) = k_1(\mathbf{x}) + zk_2(\mathbf{x})$ defines an effective local rate coefficient of infection growth, and $D_k(\mathbf{x}) = (z/4)k_2(\mathbf{x})\delta^2$ is an effective diffusion coefficient of the infection spreading, henceforth termed *epidemic* diffusion coefficient; $z$ is the number of nearest-neighbors to a node (coordination number), $z = 4, 6$ for square and triangular lattices, respectively. The diffusive-like term, in the first two lines of Eq (3), governs the diffusion of the epidemic, *not* the people. Its presence, on top the (familiar, homogeneous) infection growth rate (first term in these two lines), can describe the lateral growth of infected sub-population domains (henceforth "infected domains"), as the front—i.e. the boundary between infected and susceptible domains—propagates into susceptible domains [34, 35]. In the special case of homogeneous distribution of all populations and homogeneous rate constants, $k$ can be identified as the SEIR parameter ratio $R_0/\tau_I$, where $R_0$ is the basic reproductive rate and $\tau_I$ is the mean infectious period. It is easy to verify, by summing all lines in Eq (3), that $\partial n(\mathbf{x},t)/\partial t = 0$, as required. Thus, any initial in-homogeneous population density distribution $n(\mathbf{x})$ is not altered by our infection spreading model.

For simulation purposes we rescale the local densities by the mean total population density (in the whole region under study), $n_0$, such that $\tilde{y}(\mathbf{x}, t) \equiv y(\mathbf{x}, t)/n_0$. In particular, $\tilde{n}(\mathbf{x}) = n(\mathbf{x})/n_0$ presents the relative local population density. In addition, distance is scaled by $\delta$, i.e. $\tilde{\mathbf{x}} = \mathbf{x}/\delta$, such that $\vec{\nabla}$ becomes dimensionless. This leads to the following scaled equations

$$
\begin{aligned}
\frac{\partial \tilde{b}}{\partial t} &= k(\tilde{\mathbf{x}})\frac{\tilde{h}}{\tilde{n}}(\tilde{w}+\tilde{f}) + \frac{\tilde{h}}{\tilde{n}}\vec{\nabla}\left[\tilde{D}_k(\tilde{\mathbf{x}})\vec{\nabla}(\tilde{w}+\tilde{f})\right] - \gamma_0 \tilde{b} \\
\frac{\partial \tilde{h}}{\partial t} &= -k(\tilde{\mathbf{x}})\frac{\tilde{h}}{\tilde{n}}(\tilde{w}+\tilde{f}) - \frac{\tilde{h}}{\tilde{n}}\vec{\nabla}\left[\tilde{D}_k(\tilde{\mathbf{x}})\vec{\nabla}(\tilde{w}+\tilde{f})\right] \\
\frac{\partial \tilde{w}}{\partial t} &= \gamma_0 \tilde{b} - \gamma_1 \tilde{w} \\
\frac{\partial \tilde{f}}{\partial t} &= \gamma_1 \tilde{w} - \gamma_2 \tilde{f} \\
\frac{\partial \tilde{r}}{\partial t} &= \gamma_2 \tilde{f} \ .
\end{aligned}
\tag{4}
$$

where $\tilde{D}_k = D_k/\delta^2 = k_2$.

The parameters to be used for COVID-19 pandemic should be obtained from the up-to-date literature. Parameters that are associated with the physiological response to the disease are fairly well known, however, the basic reproductive rate, $R_0$, varies strongly between different countries due to differences in social behavior [43, 44]. In the absence of any quarantine

conditions or safety measures (e.g., use of masks), it ranges predominantly between 2 and 4 whereas initial estimates from China were $R_0 = 2$–3. In this work we chose $R_0 = 2.5$ as a typical value. It has been reported that $\tau_I = 16.6$ days [45] yielding an infection rate coefficient $k = R_0/\tau_I$ that is about $0.15 \text{days}^{-1}$. The mean time for the appearance of symptoms (from the moment of infection), $\tau_S$, is known to be about 5 days [45, 46] (ranging between 2-11 days) which sets up $\gamma_0^{-1} + \gamma_1^{-1} = \tau_S = 5$ days. In addition, there is evidence that people are infecting already about 2-3 days before showing symptoms [47], hence we set $\gamma_1^{-1} = 3$ days. It follows that the transition rate from exposed (infected but non-infecting) to presymptomatic-infecting, is $\gamma_0^{-1} \simeq 2$ days, which is quite a reasonable estimate given that viral load needs to rise before a person sheds enough virus to be considered infecting.

The rate coefficients $\gamma_1$ and $\gamma_2$, describing the transitions from presymptomatic-infecting to symptomatic-infecting, and from symptomatic-infecting to recovered, respectively, must obey $\gamma_1^{-1} + \gamma_2^{-1} = \tau_I = 16.6$ days (i.e. the whole infection period), implying $\gamma_2^{-1} = 13.6$ days. The dimensionless effective diffusion coefficient $\tilde{D}_k = k_2$ is the most difficult model parameter to estimate and is sensitive to the choice of nodes. Likely $\tilde{D}_k \ll k$, since $k = k_1 + zk_2$ and we may also assume $k_2 \lesssim k_1$. For numerical purposes in the present study we use, from now on, square lattice node geometry, i.e. $z = 4$, and choose $k_1 = k_2 = 0.03$ days$^{-1}$, yielding $k = 0.15$ days$^{-1}$ (as required) and $\tilde{D}_k = 0.03$ days$^{-1}$. Sensitivity analysis for a wide parameter range is performed in the S1 File; see Figs SI-3, SI-4, and SI-5 in S1 File. As can be expected, the speed of epidemic spread is highly sensitive to the model parameters. Yet, the spatial epidemic spreading patterns that are obtained are quite similar to one another in the range of chosen parameters, e.g., a larger $R_0$ pattern appears similar—though at shorter times—to a smaller $R_0$ pattern.

In the proceeding section we solve this spatially dependent multiple population model, at different initial conditions (using the above parameters unless otherwise stated). For a few initial conditions, we use a specific inhomogeneous density populations $n(\mathbf{x})$ to examine its effect. Obviously, to obtain realistic predictions one requires: (i) detailed local population density data (i.e. density maps), and (ii) data for the initial local densities of the above five different populations (i.e. "heat-maps"), both given to the grid size resolution $\delta$ (requiring cooperation with authorities). The present work is therefore limited to present the strength of the model and its ability to give insight on the way the infection spreads under different levels and spatial variation of quarantine or safety measures. For brevity, henceforth, we drop the '$\sim$' sign from the notations of (density) normalized spatially dependent variables, i.e. $\tilde{f} \rightarrow f$, and a similar transformation with the other variables. Recall that capital letters denote *global* quantities, i.e. spatial integrals of the lowercase spatially dependent quantities, e.g., $F = \int f(\mathbf{x}) d^2 x$, representing now the *fraction* of the specific population—out of the total population—as now $f(\mathbf{x})$ implies $\tilde{f}(\mathbf{x})$.

## Results

The initial conditions of an epidemic are unknown unless in exceptional cases, yet they have major consequences on the number of infected people. In all examples below, we use identical initial conditions for the global quantities as follows: $W = R = F = 0$ and $B = 10^{-3}$ (i.e., one out of 1000 people is in the incubation period). In our non-uniform model, we are able to analyze the effect of the different, non-uniform, initial conditions, yet identical global initial conditions. Comparing the evolution in time of the global quantities, between the locally different—yet globally identical—initial conditions, we will assess both the spreading patterns and the overall effect of the epidemic. For clarity, Table 1 summarizes the scenarios considered.

**Table 1. Summary of population distribution $n$, $b$ population at time $t = 0$, and quarantine measures employed in all figures.** For more details see the figure captions.

| Fig. | Population density $n$ | Exposed population density $b(t = 0)$ | Quarantine |
|---|---|---|---|
| 1 | Uniform | Uniform | No |
| 2 | Uniform | Two centers | No |
| 3 | Gaussian | Two centers near "city" | No |
| 4 | Uniform | Scattered centers | No |
| 5 | Gaussian | Numerous centers inside "city" | No |
| 6 | Gaussian | Scattered centers outside "city" | No |
| 7 | Nonuniform | Scattered centers outside "city" | No |
| 8 | Nonuniform | Scattered centers outside "city" | Yes, belt |
| 9 | Uniform | Numerous infections near center | No |
| 10 | Uniform | Numerous infections near center | Yes, belt |
| 11 | Uniform | Numerous infections near center | Yes, area |

## Study of model behavior

In order to better understand the basic features of the model, we commence with oversimplified cases. First, we ignore any spatial dependence of the initial ($t = 0$) sub-populations, and that of the (overall) local population density $n(\mathbf{x})$. In such cases, when the initial conditions are uniform in space, Eq (4) ensures that all sub-population densities are kept *uniform* at all times. In this limit, our model converges to a homogeneous SEPIR model (i.e. no diffusive-like terms in Eq (4)). As mentioned above, it is similar to the SEIR model with the addition of a pre-symptomatic sub-population. Although this limit cannot be achieved realistically, it is studied here for comparison with the conventional, homogeneous, models. Obviously, in this case of uniform distributions, the global (spatial integral) quantities do not provide additional information.

Fig 1 shows the model predictions for the evolution of the global variables against time $t$ when all local variables: $n$, $b$, $h$, $w$, $f$, and $r$, are spatially uniform, and for the above stated initial conditions ($W = R = F = 0$ and $B = 10^{-3}$). Fig 1(a) shows the time evolution of the model variables $H$, $B$, $W$, $F$, and $R$. At a time $t \simeq 92$ days the epidemic attains its peak, i.e. the infectious population ($W + F$) attains its maximum and later declines (dashed line). At the epidemic peak, we have $H \simeq 0.37$, i.e. the fraction of immune population is $1 - H \simeq 0.63$. Thus, within our (five sub-populations) SEPIR model, and the chosen parameters, "herd immunity" is reached at $1 - H = 0.63$ fraction of the susceptible population being infected. The fraction of recovered ("removed") people, $R$ (green curve), is increasing monotonously. In this particular example, at very long times $R$ attains a value of $R \approx 0.89$ and correspondingly $H \approx 0.11$.

A common quantity used to follow the epidemic is the accumulated number of people that have been infected until time $t$, $1 - H$. In Fig 1(b) we present $1 - H$ *vs* time on a log-log scale. We observe a separation of the growth into two (albeit very short) power-law regimes, $1 - H \simeq At^v$ (where $A$ and $v$ are numerical constants) with the early evolution exponent $v \simeq 0.26$ being much smaller than the late evolution exponent $v \simeq 4$; a smaller power-law exponent $v$ signifies a slower growth. However, given the short duration in time of these power-law regimes, it should be recognized that a power-law description is not useful for the homogeneous SEPIR model; the above analysis is shown mainly for comparison with the in-homogeneous examples below, where power-law regimes are much wider.

As a second example (Fig 2), consider a rather different initial spreading of the $b$-population, still with $n$ uniform in space. In Fig 2, and in all following figures, the epidemic heat-maps are on shown on the left-hand-side, and the corresponding global quantities are depicted

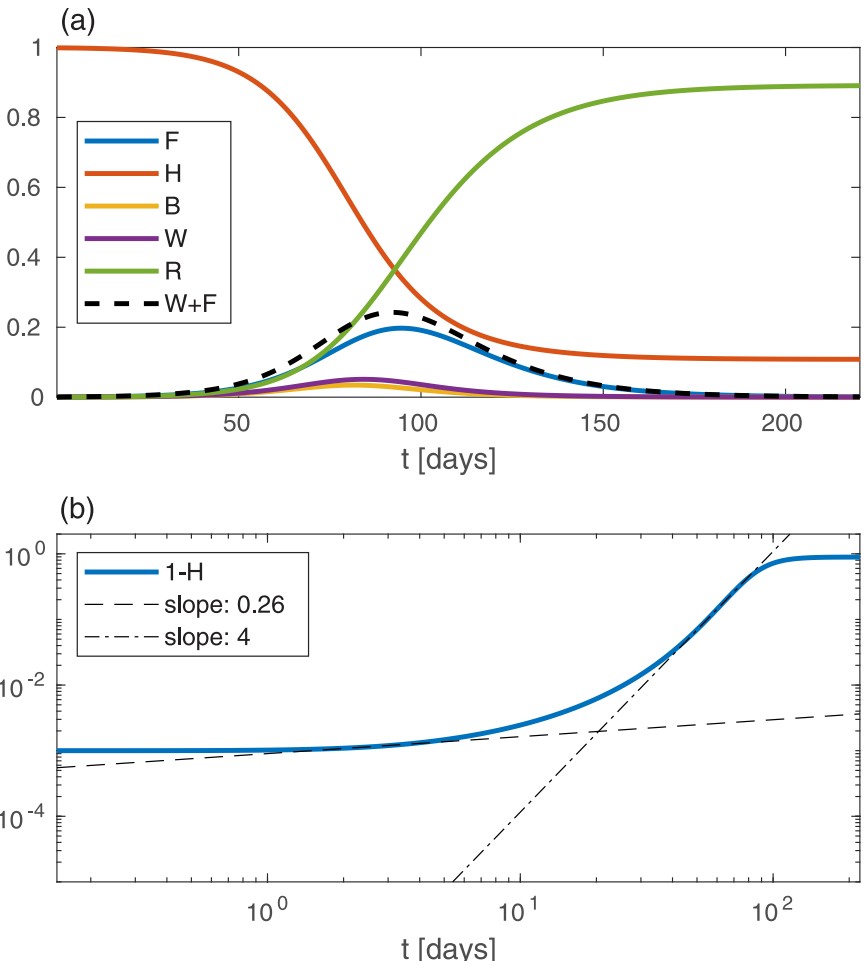

**Fig 1. Solution of the epidemic model for the case of spatially uniform population densities $n$, $b$, $h$, $w$, $f$, and $r$, against the time $t$ (in units of days).** (a) Curves depicting the global sub-population fractions (capital letters), amounting here to simple multiplication of the local densities by the area. The initial conditions are $B = 10^{-3}$ and $W = F = R = 0$. (b) A log-log plot of the cumulative infected population, $1 - H$, vs time (in days). The dashed and dash-dotted lines are fits at $t = 2$ and $t = 55$, respectively. In this and in all other figures $D_k = 0.03$, $k = 0.15$ days$^{-1}$, $\gamma_0 = 1/2$ days$^{-1}$, $\gamma_1 = 1/3$ days$^{-1}$, $\gamma_2 = 1/13.6$ days$^{-1}$.

on the right-hand-side. In Fig 2, the initial conditions are two relatively large infection centers —see panel (a). The parameters are set such that the (initial) value of $B$ is taken to be the same as in Fig 1. All other populations are initially vanishing: $w = f = r = 0$. Although $n$ is uniform at all times, all specific populations are nonuniform at $t > 0$ (even though $h$, $w$, $f$, and $r$ are set uniformly to zero at $t = 0$). The exposed sub-population centers grow and develop into ring-like structures of the symptomatic sub-population $f$, later start to overlap, and subsequently merge into one large ring that continuously spreads outwards—panels (b)-(f). The core of the rings is seen to evolve quickly to contain mostly recovered population (since $r \simeq n - f$). The evolution in time of global populations is shown in panels (g) and (h). In comparison to the case of uniform initial $b$ (Fig 1), here the peak of the epidemic occurs at much longer times ($t \simeq 324$) and is much smaller in magnitude. The early-time power-law regime quickly crosses over (at $t = 10$) to a long power-law behavior with exponent $v \simeq 2$. The latter exponent may be explained by the outward propagation of the '$f$'-rings (or those of $w + f$, not depicted in Fig 2), at constant velocity. If the front of infectious sub-populations moves at constant velocity

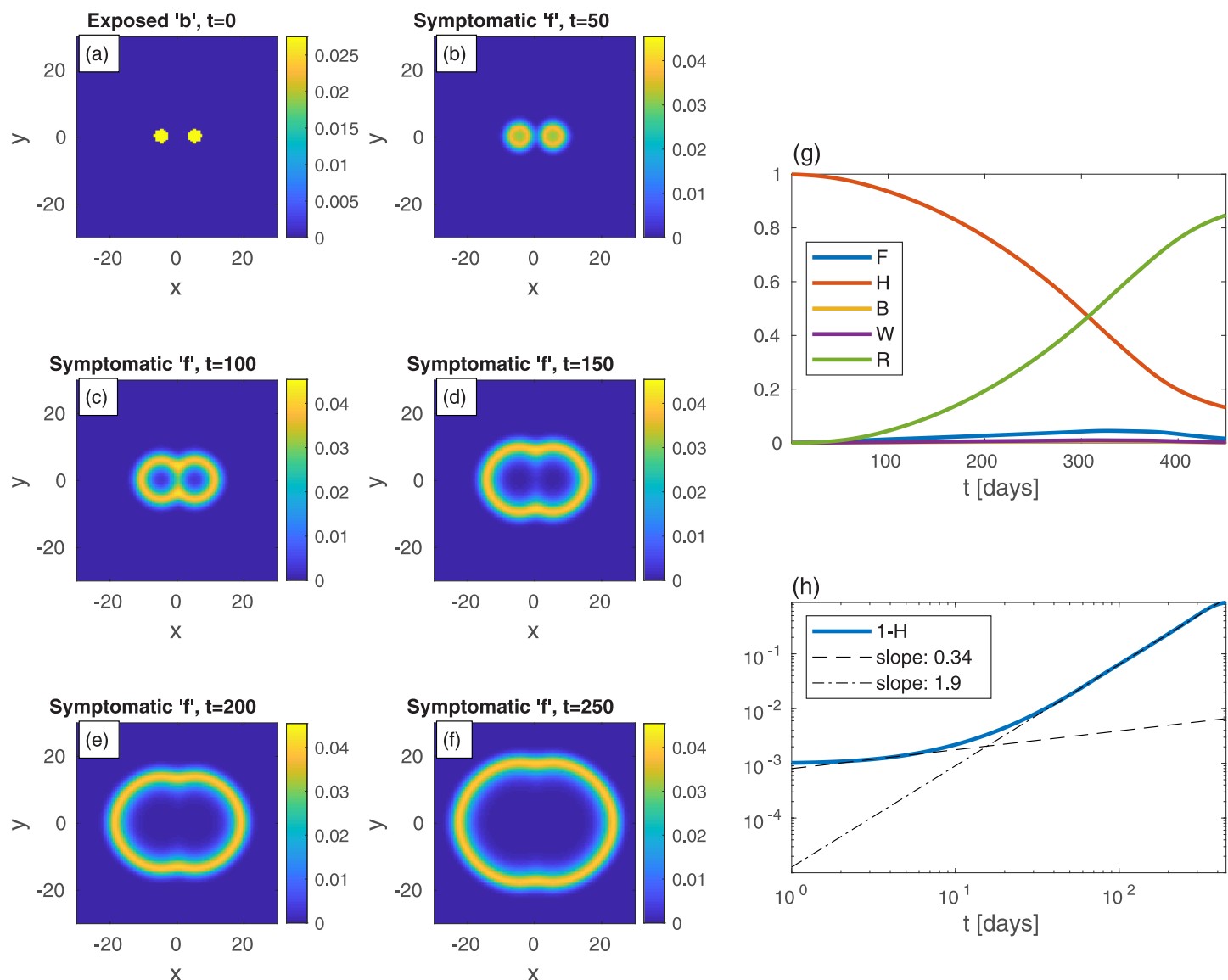

**Fig 2. Time evolution of an epidemic starting from two infection centers (in this and in all other figures, *t* is the time in days, and *x* and *y* are the spatial Cartesian coordinates).** (a) Initial conditions of *b*. *B*—the global value of *b*—is the same as in Fig 1. *n* is uniform and all other populations are initially zero: $w = f = r = 0$. Panels (b)-(f) depict the spreading pattern of the symptomatic population '*f*' as time progresses. The two circular domains grow and merge into one oval-like domain. Panel (g) shows the global sub-populations *F*, *H*, *B*, *W*, and *R vs* time *t* (in days). Panel (h) shows the cumulative fraction of infected population $1 - H$ *vs* time *t* in (in days) on a log-log scale. Compare to the cases of uniform (Fig 1). Dashed and dash-dotted curves in panel are linear fits at $t = 3$ and $t = 150$, respectively.

[34, 35], as dictated by Eq (4), it implies that the domain area, corresponding to the cumulative number of infected people, grow as $\sim t^2$.

We now turn to study a situation in which the given population density is not uniform in space, and our model is highly suitable to handle such cases. To mimic a large density variation around a densely populated area, as for a city surrounded by suburban areas, the population density is assumed to follow a centered Gaussian function with a non-zero baseline, $n(\mathbf{x}) = 10ae^{-(x^2+y^2)/\ell^2} + a$, where the standard deviation (width), $\ell$, of the density is 10, and the baseline *a* is set such that the spatial average of *n* is 0.2; see Fig SI-1 in S1 File for illustration.

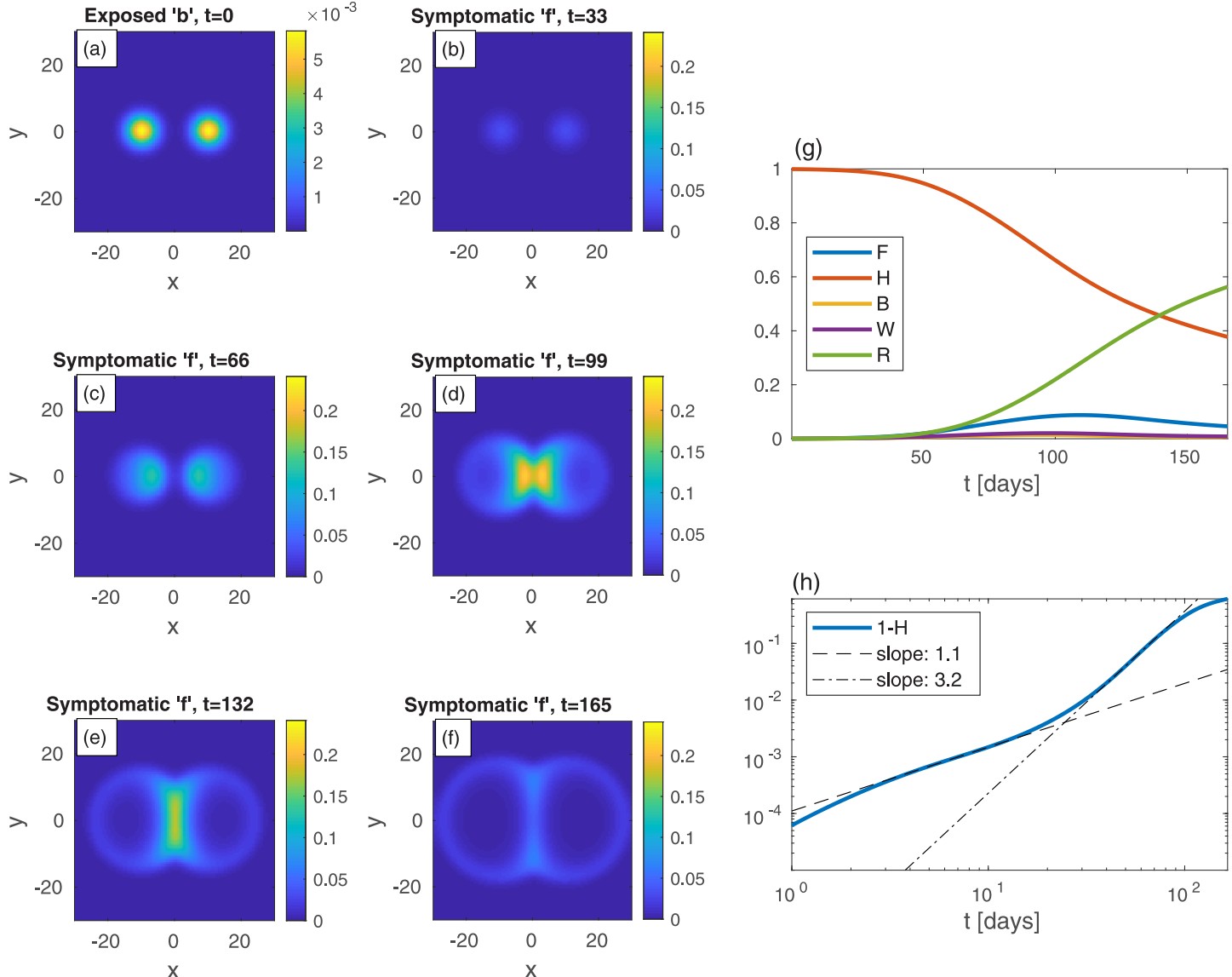

**Fig 3.** Time evolution of a epidemic starting from two infection centers near a heavily populated region, see (a) (top-left panel); $t$ is the time given in days, and $x$ and $y$ are the spatial Cartesian coordinates. $n$ is non-uniform and given by $n(\mathbf{r}) = 10ae^{-r^2/\ell^2} + a$, with $\ell = 10$ and $a$ taken such that the spatial average of $n$ is 0.2; see Fig SI-1 in S1 File for illustration. The global value of $b$ is the same as in previous figures, $B = 10^{-3}$, and all other populations are initially zero everywhere: $w = f = r = 0$. Panels (b)-(f) show the spread of the symptomatic population $f$ as time progresses. The symptomatic population quickly spreads into the denser region in the center and its density there increases dramatically. The global sub-population fractions, and the cumulative fraction of infected population $1 - H$, are shown in panels (g) and (h), respectively.

The density in the "city center", $\mathbf{x} \simeq 0$, is therefore 10 times higher than in its "suburbs", and its main core spans over a radius of 10.

Fig 3 depicts an epidemic that initiates from two infection centers of exposed ($b$) sub-population, situated on the two sides of this model "city", as in Fig 2. Interestingly, there is a remarkable strong influx of the epidemic towards the densely populated area, as evident from panels (d) and (e): the two growing infection centers merge into one large central spot with significantly more symptomatic people. This is very different from the evolution seen in Fig 2. The "late" power-law exponent governing the cumulative fraction of infected population is

$v \simeq 3.2$, i.e. in between the values obtained for the evolution depicted in Fig 2 (homogeneous $n$ only) and Fig 1 (no spatial dependence).

## More complex scenarios

Let us now look at epidemic evolution that initiates from different nonuniform initial conditions, in either uniformly or non-uniformly populated areas. Consider, first, several infection centers of the exposed sub-population $b$, randomly scattered within a *uniformly* populated area, Fig 4(a) ($t = 0$). In real-life, such initial infection centers obviously result from in-flux of

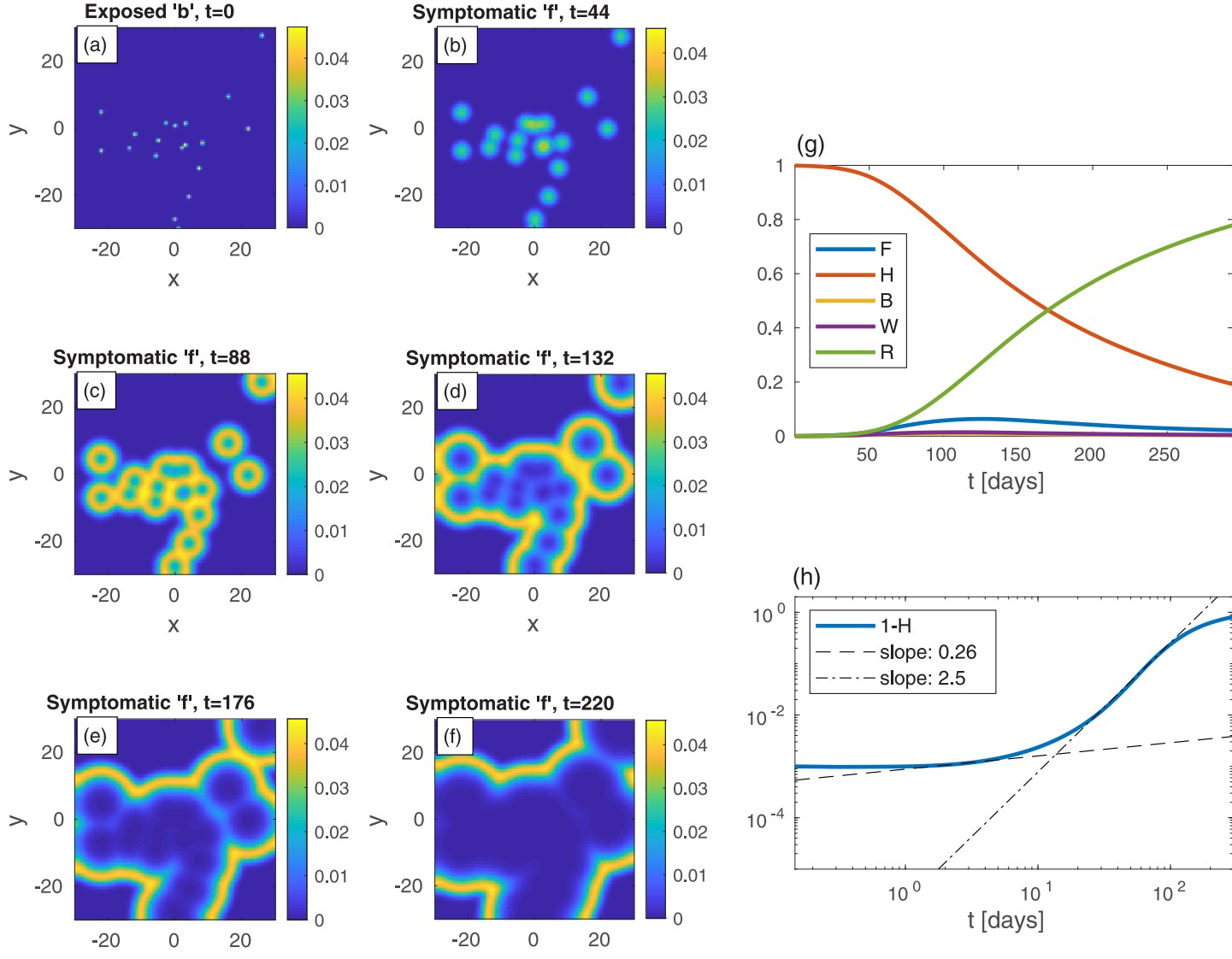

**Fig 4. Time evolution of an epidemic with randomly scattered infection centers; $t$ is the time given in days, and $x$ and $y$ are the spatial Cartesian coordinates.** (a) At $t = 0$ (top-left panel), there are small infection centers of the exposed sub-population $b$ scattered randomly in space. The global value of $b$ ($B$) is the same as in Fig 1, i.e. $B = 10^{-3}$. $n$ is uniform and all other populations are set initially to zero: $w = f = r = 0$. Panels (b)-(f) show the spread of the symptomatic population $f$ as time progresses. Panel (g) shows the global sub-populations $F$, $H$, $B$, $W$, and $R$ *vs* time $t$ (in days), and panel (h) shows the cumulative fraction of infected population $1 - H$ *vs* time in on a log-log scale. Dashed and dash-dot curves are linear fits at $t = 2$ and $t = 73$, respectively. In Fig SI-2 in S1 File we plot heat-maps of $1 - h$—the corresponding cumulative infections.

infections *via* numerous processes, e.g., when infecting people visit the city for a short period, or when infecting and susceptible people meet in different locations outside the city (creating infection events) and then return home, and so on. The global value of *b* is the same as in Fig 1, i.e. $B = 10^{-3}$, and all other populations vanish at $t = 0$ as before. Panels (b)-(f) show the time evolution of the symptomatic population *f*. As can be seen, each infection point initially grows locally and develops into a "ring" of symptomatic *f* centered around the original center (similar to Fig 2). These rings further grow and coalesce into a complex boundary pattern which keeps evolving. Panels (g)-(h) show the integrated values of *F*, *H*, *B*, *W*, and *R* and $1 - H$. In comparison to Fig 1 for the case of uniform *b*, here the epidemic evolves much more slowly, peaks at a longer time and, more importantly, the value of *F* at the peak (likewise that of $W + F$, almost indistinguishable from *F*) is significantly smaller. The "herd immunity" value deduced from the value of $1 - H$ when $W + F$ peaks (almost indistinguishable from *F*), at $t \simeq 123$, is $1 - H = 0.35$, which is quite smaller than the value obtained from the homogeneous case, Fig 1 ($1 - H = 0.63$), showing the non-universality of this value and its high sensitivity to the initial spreading conditions. The early evolution power-law exponent is similar to the uniform *b* case, but the long-time exponent $v \simeq 2.6$ is significantly smaller, leading to a significantly longer time for the epidemic to decay. In Fig SI-5 in S1 File we use identical initial conditions as in Fig 4 but with a value of $R_0$ that is twice larger—$R_0 = 5$. The snapshots in Fig SI-5 in S1 File are taken at shorter times, which yields patterns almost indistinguishable from those shown in Fig 4, and shows the universality of the epidemic spreading patterns formed by the SEPIR model.

We now go back to study situations in which the given population density is *non-uniform* in space, and consider again, as in Fig 3, a central heavily populated area ("city") whose density declines away from its center as a Gaussian with a finite width, see explanation related to Fig 3. Here, however, we consider more realistic initial conditions, where exposed (*b*) sub-population centers are scattered in a number of places. We distinguish here between two different cases: (i) The infection (*b* sub-population) initiates from multiple centers within the city (Fig 5). (ii) The infection evolves from several, randomly distributed, centers in the city periphery (Fig 6).

Consider first an infection initiating from around the city center (case (i)), as several small *b*-centers randomly distributed within the city core. Initially, the epidemic consumes non-negligible portion the susceptible (*h*) sub-population within the city core, associated with a substantial growth of *f*, see Fig 5(b) and 5(c). After this early evolution ($t \gtrsim 90$) the infection slowly spreads outward by formation of ring-like patterns, bearing some similarity to the above studies. Conversely, when the infection initiates from the city outskirts (case (ii), Fig 6), the pattern is more heterogeneous, and local infection centers grow effectively independently of each other. Here, as the epidemic reaches the core of the city, the relatively high density of susceptible population (*h*) allows the *f*-population to keep rising. Hence *F(t)*, seen in case (ii) (Fig 6(g)), grows for a somewhat longer time and reaches a higher peak, as compared to case (i) (Fig 5(g)). The apparent "herd immunity" value ($1 - H$ at the peak of *F* or $W + F$) is also higher in case (ii) ($1 - H \simeq 0.45$) than in case (i) ($1 - H \simeq 0.27$), and both values are lower than the one obtained in the homogeneous case (Fig 1), demonstrating again the sensitivity of this value to both initial conditions and spatial density variation.

## Quarantine strategies

So far we looked at the unperturbed spread of an epidemic. Yet, authorities often use numerous active tools to confine the disease or slow down its spread. Usually people are instructed to stay home for a considerable period, the so-called "quarantine", "lockdown" or "stay-at-home"

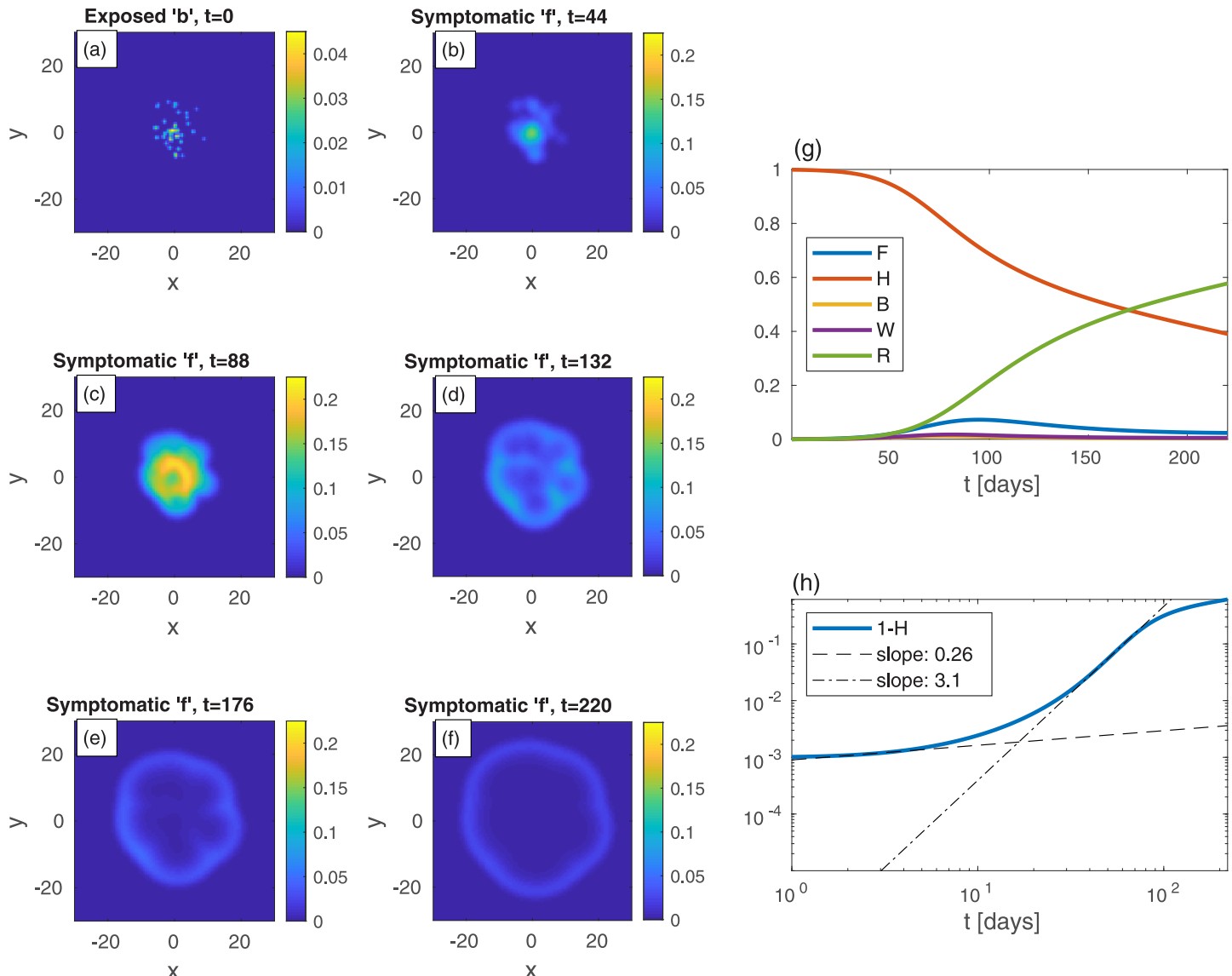

**Fig 5.** Time evolution of an epidemic starting from multiple infection centers inside a heavily populated region ("city", panel (a)); $t$ is the time given in days, and $x$ and $y$ are the spatial Cartesian coordinates. The population density of the city $n$ is nonuniform and given by $n(\mathbf{x}) = 10ae^{-x^2/\ell^2} + a$, with $\mathbf{x} = (x,y)$, $\ell = 10$, and $a$ taken such that the spatial average of $n$ is 0.2; see Fig SI-1 in S1 File for illustration. The integrated initial value of $b$ ($B = 10^{-3}$) is the same as in all previous figures. All other populations are initially zero: $w = f = r = 0$. Panels (b)-(f) show the spread of the symptomatic sub-population $f$ as time progresses. The global sub-populations and the cumulative infected population $1 - H$ are shown in panels (g) and (h), respectively.

order. In addition, roads connecting between "hotspots" and uninfected regions are sometimes blocked. Here we consider two types of quarantine strategies. We define local "area lockdown" as strong restrictions on activity within a certain area. Examples where "area lockdown" have been employed are: Wuhan (China), Anxin county (China), dozens of residential compounds in Beijing (China), Bnei-Brak (Israel), New York (USA)—"stay-at-home" order [48, 49], Florida State—"stay-at-home" order [50, 51]. We also propose another strategy that inflicts less burden on the society, which we term "belt quarantine". Here movement restrictions are imposed between a certain region and its surroundings, but not inside the region itself. Partial belt quarantine has been employed for instance in Canada, as the whole state reopened from

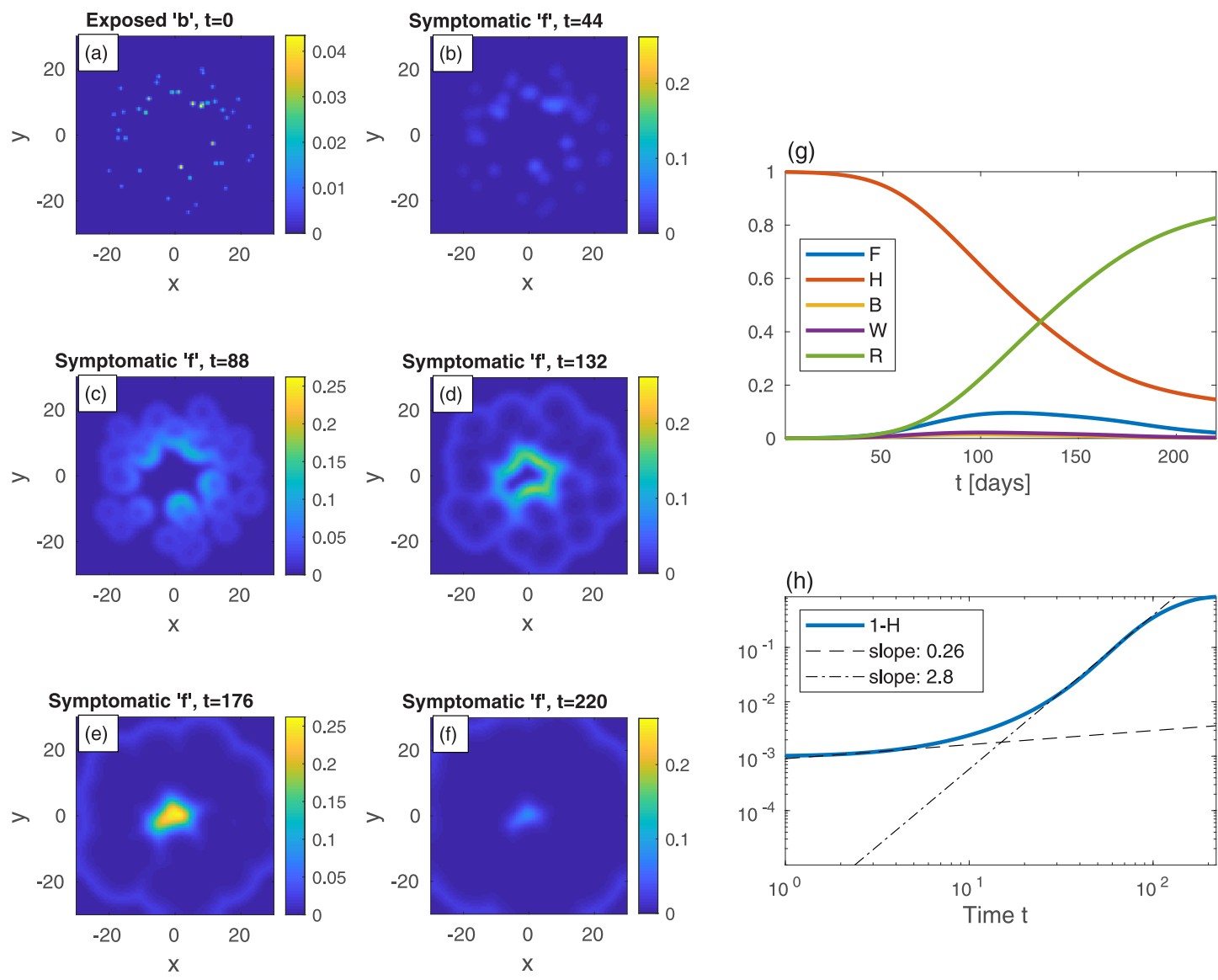

**Fig 6. Same as in Fig 5 but with the initial ($t = 0$) infection centers located outside of the "city".**

lockdown, using checkpoints and roadblocks [52–54]. In both, there are fewer and less frequent encounters between people implying locally reduced values of $D_k$ and $k$.

We commence by examining the effectiveness of belt quarantine that is imposed in order to protect an uninfected heavily populated area ("city") from its infected surroundings. For comparison, we also consider the case *without* quarantine, see Fig 7. To mimic belt quarantine, see Fig 8, we impose significantly reduced values of $D_k$ and $k$ within a belt surrounding a city. Specifically, between radii 10 and 12, the values of $D_k$ and $k$ are reduced to 20% of their background values (that are identical to those used for Fig (7)). The density profile (in both figures), describing the city and its surroundings, is *not* a Gaussian (as in the previous examples), but uniform within the city, and uniform but *ten-folds lower* outside the city. The initial ($t = 0$) scattered centers that surround the city, of exposed sub-population ($b$), are identical in both figures (panel (a) in both). The resulting epidemic spreading patterns shown in

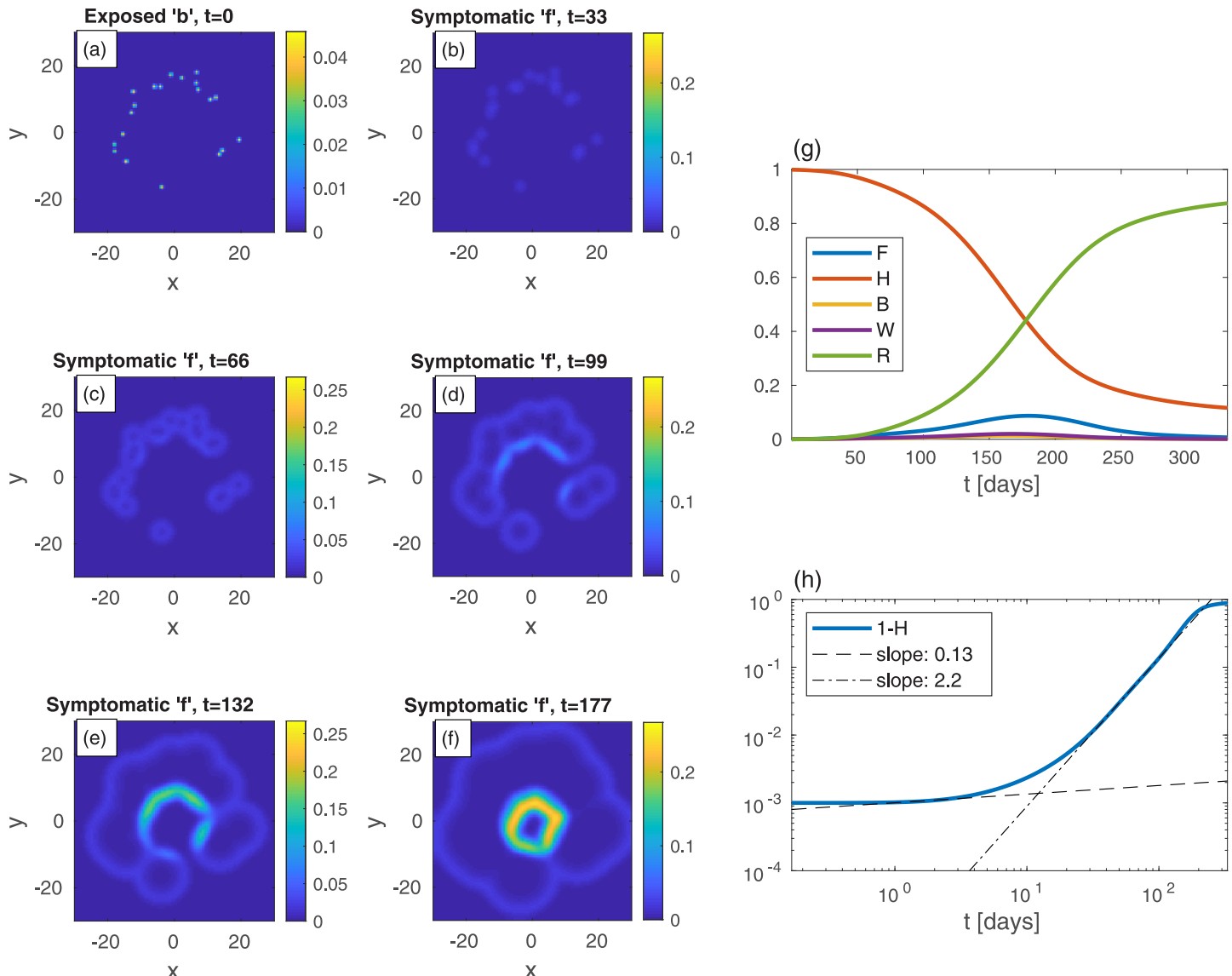

**Fig 7.** Time evolution of an epidemic starting from multiple random infection centers near a heavily populated region—mimicking a city—see (a) (top-left panel); $t$ is the time given in days, and $x$ and $y$ are the spatial Cartesian coordinates. $n$ is non-uniform, and is given by $n = 10a$ within a circle of radius 10, and $n = a$ outside of that circle, with $a$ taken such that the spatial average of $n$ is 0.2. The initial ($t = 0$) integrated value of $b$ is the same as in previous figures, $B = 10^{-3}$. All other populations are initially zero: $w = f = r = 0$. The global sub-population fractions are shown in panels (g)-(h).

Figs 7(b)–7(f)) and 8(b)–8(f), suggest that with the protective belt quarantine the infection takes considerably longer time to invade the city.

Comparing the resulting evolution in time of the global sub-populations, Fig 7(g) and 7(h) vs Fig 8(g) and 8(h), we observe that the belt quarantine slows down significantly the growth of symptomatic population $F$. Under quarantine (Fig 8(g)), $F$ first develops a very wide plateau corresponding to the epidemic spreading only in the surroundings, outside the quarantined city. Later, at $t \simeq 230$ days, $F$ further grows and builds a major (second) peak ($t \simeq 268$), corresponding to the epidemic spreading in the quarantined zone, Fig 8(f). While the height of the peak without quarantine is similar to the one with quarantine, the first occurs at about 90 days earlier ($t \simeq 177$ days). As a result, the value of $H$ at the major epidemic peak is, in fact, smaller

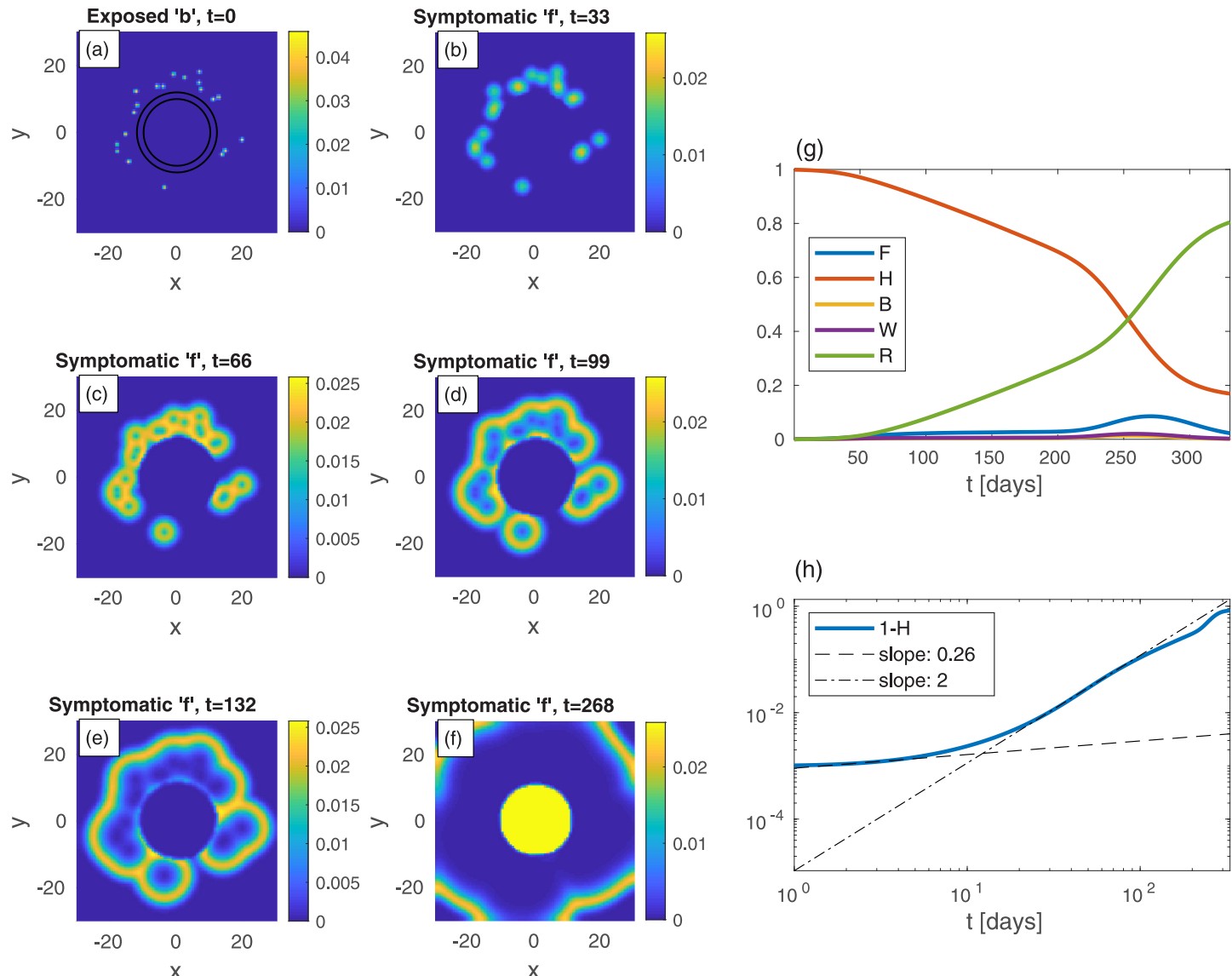

**Fig 8. Belt quarantine.** Time evolution of an epidemic starting from multiple random infection centers (see panel (a)), near a city *identical* to that of Fig 7: $n$ is nonuniform, and is given by $n = 10a$ within a circle of radius 10, and $n = a$ outside of that circle, with $a$ taken such that the spatial average of $n$ is 0.2; $t$ is the time given in days, and $x$ and $y$ are the spatial Cartesian coordinates. The "city" is under a protective circumferential "belt", formed by two concentric circles (radii 10 and 12), within which $D_k$ and $k$ are reduced to 20% of their values elsewhere. The initial ($t = 0$) integrated value of $b$ is the same as in previous figures, $B = 10^{-3}$. All other populations are initially zero: $w = f = r = 0$. Panels (b)-(f): The infection is seen to spread quickly within the external area, but penetrates very slowly into the protected region. The global sub-population fractions are shown in panels (g)-(h); $F$ shows a very wide plateau followed by a higher peak.

in the presence of quarantine, as most of the (whole) region has been infected before penetration to the quarantined city occurred. This suggests that a belt quarantine, protecting a highly populated area from its surrounding, only delays the epidemic spread and, in case the majority of population lives in the protected zone, is unable to flatten the epidemic curve significantly.

Finally, we turn to investigate a neighborhood, which is considered as the epidemic epicenter, within a large, uniformly populated, urban area. The population density $n$ is thus uniform in the whole region of study. The neighborhood area is assumed to be a circle of radius 11, and within it there is initially a high fraction of exposed population $b$ as multiple, randomly scattered, centers. We consider again belt quarantine which is imposed on the neighborhood in

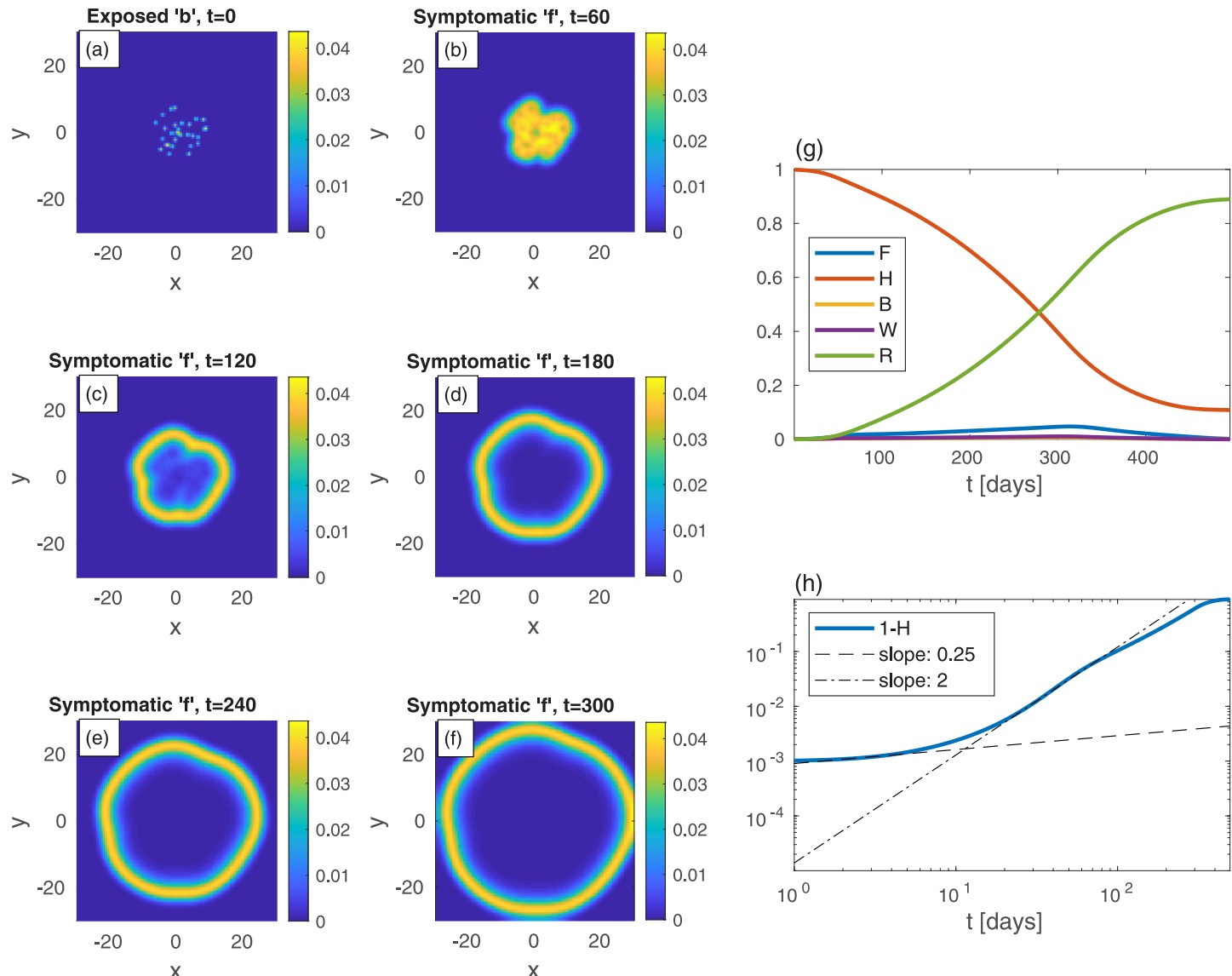

**Fig 9.** (a)-(f): Time evolution of a epidemic starting from multiple random infection centers with uniform $n$, see (a); $t$ is the time given in days, and $x$ and $y$ are the spatial Cartesian coordinates. The global value of $b$ is the same as in previous figures, $B = 10^{-3}$. All other populations are initially zero: $w = f = r = 0$. Both $D_k$ and $k$ are uniform. The symptomatic population $f$ spreads and forms ring-like structures that expands in time. Panels (g) and (h) show (respectively) the different global populations and the cumulative infected population, $1 − H$, $vs$ time $t$.

order to contain the epidemic within it. In addition, we examine the effect of a more severe measure: lockdown on the whole neighborhood, which we term "area lockdown". Fig 9 depicts the spreading patterns in the *absence* of quarantine, which is shown for comparison. Fig 10 represents the effect of *belt quarantine* between radii 10 and 11. Fig 11 depicts the consequences of *area lockdown* within the whole (highly infected) neighborhood. In both belt quarantine and area lockdown the values of $D_k$ and $k$ are reduced to 20% of their background values.

The comparison of the epidemic spreading patterns between the above three cases, shows that with belt quarantine (Fig 10) the escape of infection from the neighborhood to its surroundings takes longer time relative to the no quarantine situation (Fig 9). Note that area

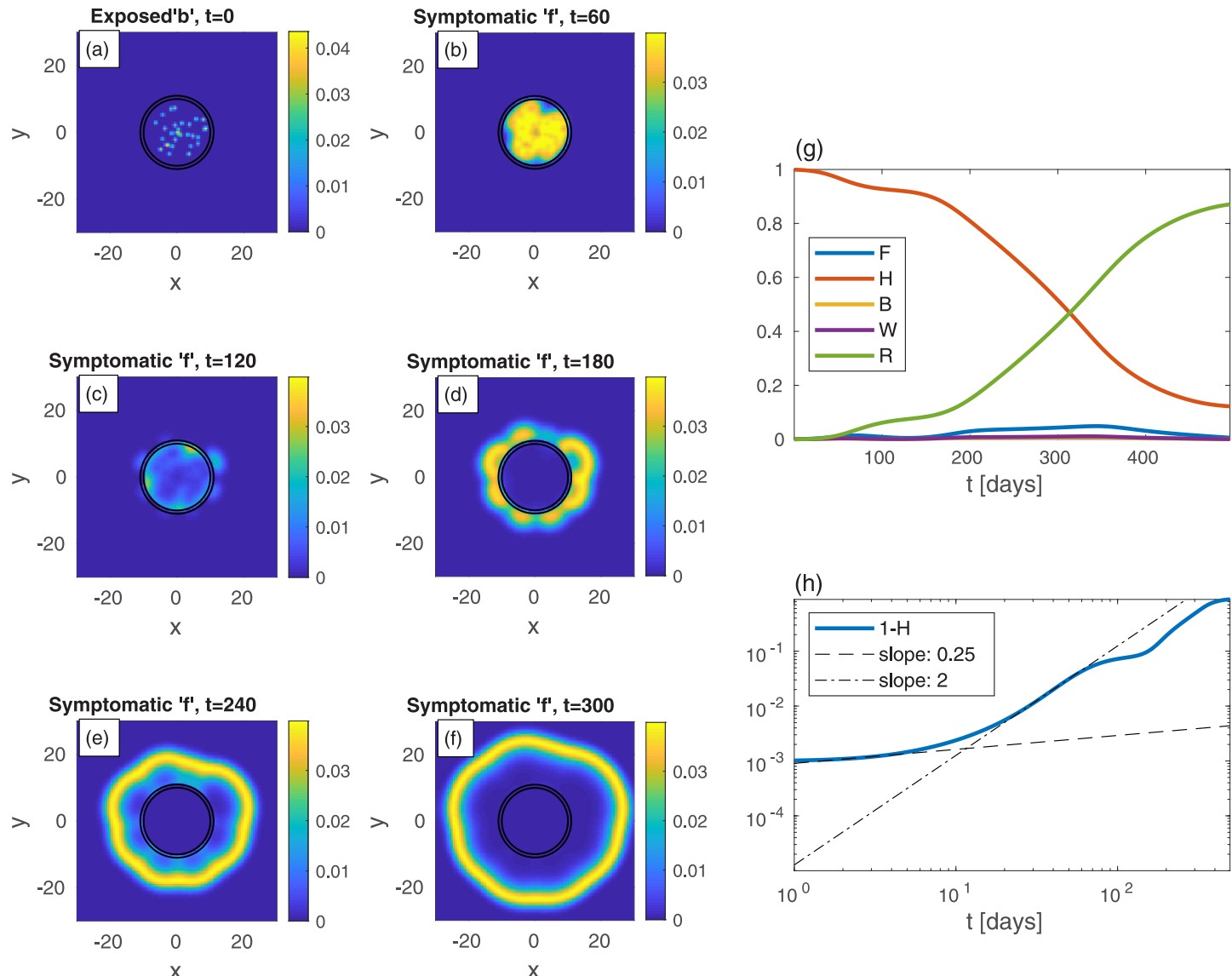

**Fig 10. Belt quarantine.** Same as in Fig 9 but now with a protective "belt" quarantine in the region between the two concentric circles (radii 10 and 11). Within the belt, the values of $D_k$ and $k$ are reduced to 20% of their values in the rest of the region. Initially the epidemic is confined to the quarantined region, but at long times it leaks out through the belt and contaminates the exterior. Note the relatively isotropic spreading patterns in the exterior, despite the initial non-isotropic $b$ depicted in (a).

lockdown (Fig 11) is even more efficient than belt quarantine in prolonging this escape time. Furthermore, it appears that in the case of belt quarantine the late-time spreading pattern are nearly isotropic, while in the case of area lockdown the escape pattern is highly non-isotropic. This occurs since prior to the escape, in the case of belt quarantine the infected population homogenizes rather quickly within the neighborhood, while for area lockdown the slow spreading within the neighborhood prevents this homogenization.

The overall effect of the different quarantine measures on the global populations is shown in panels (g) and (h) of the respective figures. Belt quarantine produces two peaks of symptomatic population (*F*), one (small) that corresponds to the spreading within the neighborhood, and the other (larger) corresponding to the external spreading after escape from the neighborhood occurred. The major rise in *F* is delayed mostly under area lockdown, due to the relative

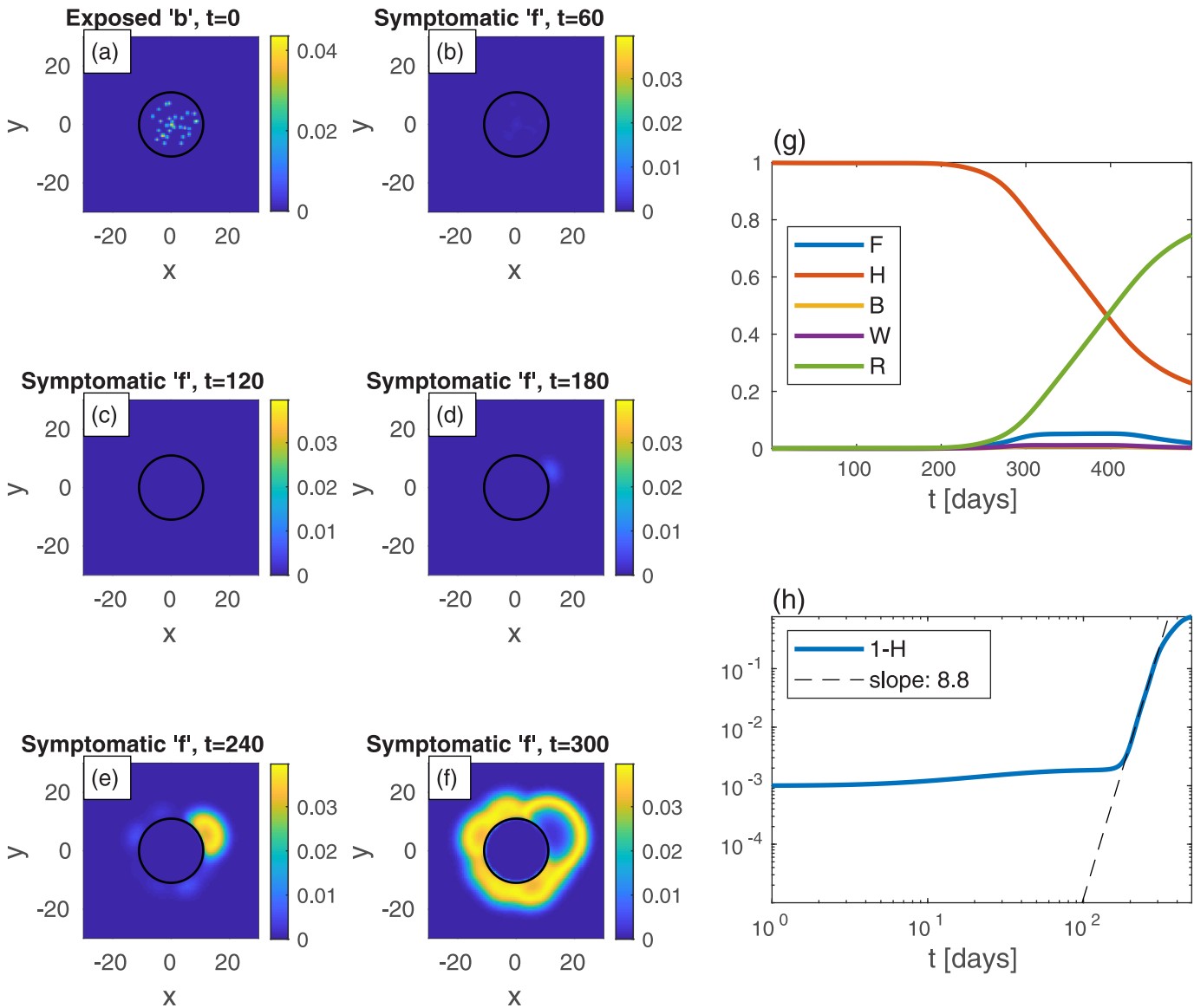

**Fig 11. Area lockdown.** Same as Fig 9 but now the quarantine is throughout the whole area within a circle of radius 11. Within this quarantined area, the values of $D_k$ and $k$ are reduced to 20% of their values in the rest of the region. The "contamination" of the exterior is slower than in Fig 10. Also note the relatively anisotropic spreading patterns seen at long times in the exterior, as compared to Figs (9) and (10).

long epidemic escape time from the neighborhood. Another pronounced effect of area lockdown appears in the slow increase of cumulative fraction of infected population, $1 - H$: at the longest simulation time we obtain $1 - H \simeq 89\%$ without quarantine, $1 - H \simeq 88\%$ with belt quarantine, and $1 - H \simeq 77\%$ with area lockdown. Thus, it is clear that area lockdown should be the preferred choice for containing the epidemic; belt quarantine, unless almost hermetic ("leakage-proof"), has a minor contribution. Moreover, none of the two quarantine strategies considered is able to reduce significantly the *level* of the (major) peak in $F$ (even though its timing is delayed), which is considered important for the ability of health systems to cope with the epidemic. This occurs due the escape of the epidemic from the quarantined region, leading to its free spreading. This may be prevented by moving the quarantine to newly infected regions,

which has not been addressed here. If such a careful orchestration is performed, localized quarantine strategies can have a strong impact.

## Conclusions

A general mathematical framework is presented for the spatial spreading of an infectious disease. We take into account nearest-neighbor node infection kinetics, and show that it leads to a diffusion-like term in the dynamical equations, thereby providing a unified framework for heterogeneous spread of the epidemic. Nodes are defined by the assumption that within each node the frequency of contacts is still uniform, thus following a homogeneous model description. This allows to estimate the lateral (linear) size of a node $\delta$ (i.e. area equal to Â $\delta^2$), for example by examining the mean distance traveled by people—e.g., 30 km under moderate restrictions or normal life, or 1 km under strong lockdown—and makes our model flexible enough to describe infectious disease spreading on a scale ranging from a neighborhood to a whole country.

We focus here on an epidemiological spreading model for COVID-19 with inherent spatial dependency of five populations, the in-homogeneous SEPIR model. We show that the complex pattern formation is sensitive to the initial conditions, i.e., to the spatial location of the exposed population, which has important consequences for the total number of infected people. Our general mathematical framework allows to include many more populations than those appearing in our SEPIR model, as suggested by other studies [19, 20, 22]. In particular, the asymptomatic population forms about 16%-40% of the total infectious population [55], and its infectious properties are quite different from those of the symptomatic population [56]. As knowledge is accumulating, it would be worthwhile to include into the model the asymptomatics, either as a single population, or as two separate populations, "asymptomatic-normal" and "asymptomatic-super-spreaders".

The homogeneous (i.e. prefect mixing) models cannot provide epidemiological heat-maps, such as those occasionally appearing on-line (e.g., for South Carolina [37]), and prediction of such heat-maps is the main purpose of our work. Moreover, failures of the homogeneous models might be due to their inability to account for the spatial spreading of the epidemic, which yields an overestimation of the epidemic growth rate. Hence, observed deviations from the homogeneous models, i.e. as effective power-law regimes [26, 30], can be rationalized without assuming time variation of the infection rates, as is customary done in practice. We show this in the present work by comparing the results of the homogeneous SEPIR model (Fig 1) to the results of our *in*homogeneous SEPIR model (Figs 2–6). It is gratifying that some of the power-law exponents found in our studied examples are close to observed exponents in different counties; in particular the exponent $v \simeq 2$ has been deduced for China and Iran [30].

The evolution of the homogeneous SEPIR model suggests "herd immunity" at about $H \simeq 0.37$ (corresponding to $W + F$ reaching its maximum, Fig 1)—i.e. fraction of immune population $1 - H \simeq 0.63$—indeed very close to the known SIR result $1 - H = 1 - (1/R_0) = 0.6$ for $R_0 = 2.5$. Importantly, the "herd immunity" values that could have been (wrongly) inferred from the *in*homogeneous evolution, i.e. assuming the homogeneous SEPIR model to still hold, are much lower. For example, from Fig 4 one obtains $1 - H \simeq 0.35$. Moreover, different initial conditions are seen to lead to different *apparent* herd immunity values. Thus, interpreting observed epidemic curves based on homogeneous models may lead to wrong conclusions regarding the population reaching herd immunity. Further support for this conclusion has been reached by examining epidemic curves of European countries, where it was noted that spatial heterogeneity can lower the apparent herd immunity value [57], supporting our model conclusions.

Our model can naturally describe the flux of infection from a suburban area into a densely populated city, or in the opposite direction. Interestingly, we find that relative "curve flattening" of the infected-symptomatic population ($F$) can naturally occur due to either non-uniform population density, non-uniform distribution of initial infectious populations, or the combination of both. This may have important implications when comparing the epidemic evolution in different regions or states, where one needs to distinguish between the effects of quarantine measures and population conditions. For example, when comparing Sweden (essentially no quarantine measures) and Israel (severe quarantine measures), conclusions might be hampered.

Accurate predictions of COVID-19 heat-maps will require complete data sets for: (i) the geographic (i.e. position dependent) overall population density $n(\mathbf{x})$, and (ii) the "initial heat-maps" for the different five population types defined in our model. In many countries $n(\mathbf{x})$ can be obtained from public resources, e.g., see [58]. Unfortunately, the initial heat-maps, i.e. initial conditions for the five population densities, are currently publicly available only for a small number of countries (apparently due to privacy regulations), and moreover, usually limited only to the time-cumulative density of the infected population, i.e. $1 - h(\mathbf{x})$ [37]. Cooperation with health authorities is needed to obtain complete data sets. Our predicted evolution of COVID-19 heat-maps (see Fig SI-2 in S1 File for the cumulative infected population, corresponding to the heat-maps shown in Fig 4) show strong resemblance to those available on public resources [37]; future work will be devoted to quantitative comparison. Incomplete testing data is not going to severely hamper our predictions so long as it is uniform in space (e.g., only 10% of the COVID-19 positives are detected everywhere), and in general we may expect this to be so within a certain country where a uniform testing policy is adopted. Obviously, absolute predicted numbers will not be obtained without the use of an adjusting factor between tested and predicted numbers. It should be noted that when normal life conditions are restored, our model has to be modified to include far-distance traveling; this can be readily done using long distance infections, which we defer to future work.

Importantly, the possibility to mimic in our model spatially-varying and evolving quarantine or lockdown conditions, by using both spatially-dependent (as done in this work) and time-dependent values of $D_k$ and $k$, will allow a quantitative predictive tool for the effectiveness of quarantine measures. For instance, in Israel, a plan has been proposed ("the traffic lights plan" [59]) to impose differential lockdown measures on cities with strong outbreak ("red cities"), and similar plans have been issued for the UK [60]. Our model allows to simulate the impact of these measures and make comparison with the evolution without intervention. We hope authorities will use this tool, in addition to established venues [8, 61], to simulate different lockdown policies for choosing the best exit strategy [62].

## Supporting information

**S1 File.**
(PDF)

## Acknowledgments

We are grateful to Ariel Kushmaro for insightful discussions.

## Author Contributions

**Conceptualization:** Yoav Tsori, Rony Granek.

**Data curation:** Yoav Tsori, Rony Granek.

**Formal analysis:** Yoav Tsori, Rony Granek.

**Funding acquisition:** Yoav Tsori, Rony Granek.

**Investigation:** Yoav Tsori, Rony Granek.

**Methodology:** Yoav Tsori, Rony Granek.

**Project administration:** Yoav Tsori, Rony Granek.

**Resources:** Yoav Tsori, Rony Granek.

**Software:** Yoav Tsori, Rony Granek.

**Supervision:** Yoav Tsori, Rony Granek.

**Validation:** Yoav Tsori, Rony Granek.

**Visualization:** Yoav Tsori, Rony Granek.

**Writing – original draft:** Yoav Tsori, Rony Granek.

**Writing – review & editing:** Yoav Tsori, Rony Granek.

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
