## [Decision Letter · Decision Letter 0]

2 Sep 2020

PONE-D-20-21655

Epidemiological model for the inhomogeneous spatial spreading of COVID-19 and other diseases

PLOS ONE

Dear Dr. Tsori,

Thank you for submitting your manuscript to PLOS ONE. After careful consideration, we feel that it has merit but does not fully meet PLOS ONE’s publication criteria as it currently stands. Therefore, we invite you to submit a revised version of the manuscript that addresses the points raised during the review process.

We look forward to receiving your revised manuscript.

Kind regards,

Emanuele Giorgi

Academic Editor

PLOS ONE

Journal Requirements:

2.

We suggest you thoroughly copyedit your manuscript for language usage, spelling, and grammar. If you do not know anyone who can help you do this, you may wish to consider employing a professional scientific editing service.  

Reviewers' comments:

Reviewer's Responses to Questions

**Comments to the Author**

1. Is the manuscript technically sound, and do the data support the conclusions?

Reviewer #1: Partly

Reviewer #2: Partly

Reviewer #3: Yes

2. Has the statistical analysis been performed appropriately and rigorously? 

Reviewer #1: I Don't Know

Reviewer #2: I Don't Know

Reviewer #3: I Don't Know

3. Have the authors made all data underlying the findings in their manuscript fully available?

Reviewer #1: Yes

Reviewer #2: Yes

Reviewer #3: Yes

4. Is the manuscript presented in an intelligible fashion and written in standard English?

Reviewer #1: Yes

Reviewer #2: No

Reviewer #3: Yes

5. Review Comments to the Author

Reviewer #1: Please see attached comments.

Responses to answers to questions justified:

Is the manuscript technically sound, and do the data support the conclusions? PARTLY

I answered PARTLY because I think the model is technically sound, but the conclusion demonstration why this framework is better for COVID-19 is lacking

Has the statistical analysis been performed appropriately and rigorously? I DONT KNOW

I find the derivation of the model itself rigorous. I answered I DONT KNOW because I would like justification on why the assumption that individuals cannot infect others beyond their nearest neighbor is appropriate.

Reviewer #2: Review Comments

The authors have presented an interesting mathematical model for COVID-19 that accounts for spatial heterogeneity in population density and population mixing. I believe this type of model can be very useful and an improves upon the SEIR model by representing the spatial dynamics of transmission. However, the manuscript is written in a very technical manner and is not accessible to the general readership of PLOS ONE. In order for this work to be published in this journal, I suggest it be rewritten to be more explicitly relevant to the current COVID-19 pandemic. I have provided specific comments below, but I broadly recommend that the authors better describe how their model and findings improve our understanding of COVID-19 transmission dynamics and relevant interventions. How does the model improve upon other existing models that account for spatial heterogeneity in population density and mixing? How do the specific simulations presented here relate to the real world spread of COVID-19? How plausible is it that the data needed for this model will be available? What are other possible uses of this model in the context of the COVID-19 pandemic?

Introduction

- Lines 32-38: These sentences are unclear and have a lot of terminology that has not been defined. Could you rewrite them defining terms such as power-law growth and “front propagation”. As it stands, I’m unsure what the hypothesis is and how it relates to the spatial mixing discussed above.

- Lines 49 – 53: These sentences don’t say much about the results and seem to be saying that the model is indeed different from SEIR, which is true by construction. If the authors want to include a sentence describing their results, it should be more specific and describe what is learned from their model in the larger context of transmission dynamics and interventions.

- It would be useful to add a section to the introduction about other models that have attempted to account for spatial heterogeneity, quarantine, and heterogeneous mixing. This will allow the authors to make more explicit how their model is novel and/or adds to the existing literature.

Methods:

- Line 115: This section should be updated to include more recent literature estimating these parameters across various populations and time periods. There have been a myriad of studies that estimate these parameters and the authors should provide justification for their parameter value choices.

- Line 124-125: These parameter values are very uncertain, so the authors should use sensitivity analyses or multiple simulations to explore how sensitive the results are to the chosen values.

- Line 138: I wouldn’t think that the “heat map” data the authors are describing is readily available in many contexts. They should discuss the availability of such data here and in the conclusion, and provide citations and examples of these data, if available.

- The authors should make it clearer in the introduction that they are demonstrating the use of a new model structure but are not relating it to any real population. That said, why are the authors not focusing on a specific region or population affected by COVID-19?

Results

- Generally, the results should be written more for a general audience. For each simulation, the authors should make it clear why the examples were chosen and how they relate to real-world transmission situations. What do the results of each simulation tell us about COVID-19 transmission? For example, what would cause each of these simulation scenarios to happen in the real world? Have there been examples of such transmission scenarios the authors can point to? What are the implications of the initial conditions of the simulations for how we interpret the results?

- All heatmap figures are hard to understand due to a lack of legend and labeling. The axes (currently X and Y) of each panel need to be labeled and the color scheme needs to be defined and labeled.

- If the authors want to do simulations in a city, why not use real population density data from a relevant city experiencing a COVID-19 outbreak?

- I think the city analyses are useful and more relevant to real life, but the decisions about how to initiate infections should be explained more in terms of real-life situations. How would these initial conditions occur in the real world?

- Are “belt quarantine” and “area lockdown” real strategies used to fight the spread of COVID-19? Please provide examples and citations for these strategies. If they are not common strategies, can the authors include simulations of more commonly used quarantine strategies?

Conclusion

- This section needs more discussion of existing literature and models that account for spatial heterogeneity and estimate the impacts of quarantine on transmission. How do the results from this model/analysis compare with existing literature? How does this paper add to our understanding of quarantine interventions and transmission dynamics?

- As mentioned above, the authors need to add more in-depth discussion of the data needed for this model to be operationalized. I’m not familiar with what they call “heat maps” and am not sure how available they will be. Please add some information about what exactly these data are and where they are available.

- It would also be useful for the authors to more explicitly discuss the potential uses of their model in the context of the COVID-19 pandemic.

Minor comment: The authors should edit the writing to make sure the grammar is correct throughout the paper.

Reviewer #3: PONE-D-20-21655

Page 2 Lines 8-10: "Infectious disease spreading models are largely based on the assumption of perfect and continuous \\stirring", similar to the one used to describe the kinetics of spatially-uniform chemical reactions. In particular, the well-known susceptible-exposed-infectious-recovered (SEIR) model, builds on this assumption."

This is true in some cases, but is very limited. The term SEIR just refers to the compartments which are used, and many extensions are available in both the deterministic and stochastic literature (the authors note a few). It would be clearer to call this the homogenous-mixing or baseline SEIR model.

It seems that more information about how the proposed techniques differ from other deterministic approaches which appear superficially similar. A few examples:

https://aip.scitation.org/doi/full/10.1063/1.5116807

https://link.springer.com/article/10.1007/s11587-013-0151-y

https://arxiv.org/abs/2005.03499

> Page 2 line 31 "often require refitting the infection rate constant as the epidemic progresses"

I don't believe this should neccessarily be listed as a negative - in real epidemics/pandemics, there is usually very little reason to believe that epidemic intensity (which involves infectiousness as well as contact rates) is constant over space or time. The authors later note this and express the idea more clearly, but I think this could be clarified.

I think the manuscript would be greatly enhanced by a more detailed explanation of the degree of model fit, and any appropriate sensitivity to the driving parameters. In addition, the discussion should address the following issues:

1. Application of this technique to areas of sparse spread/inflated zero counts

2. Limitations concerning the use of incomplete testing data for validation

3. People travel great distances, which would seem to limit the utility of this model. Can it be applied to large areas where non-contiguous spreading events occur, or would that require generalization?

6. PLOS authors have the option to publish the peer review history of their article (what does this mean?). If published, this will include your full peer review and any attached files.

Reviewer #1: No

Reviewer #2: No

Reviewer #3: No

---

## [Author Response · Author response to Decision Letter 0]

28 Oct 2020

Please see attached file `response_to_reviewers.docx' with detailed response to the reviewers.

---

## [Decision Letter · Decision Letter 1]

2 Dec 2020

PONE-D-20-21655R1

Epidemiological model for the inhomogeneous spatial spreading of COVID-19 and other diseases

PLOS ONE

Dear Dr. Tsori,

Thank you for submitting your manuscript to PLOS ONE. After careful consideration, we feel that it has merit but does not fully meet PLOS ONE’s publication criteria as it currently stands. Therefore, we invite you to submit a revised version of the manuscript that addresses the points raised during the review process.

We look forward to receiving your revised manuscript.

Kind regards,

Emanuele Giorgi

Academic Editor

PLOS ONE

Reviewers' comments:

Reviewer's Responses to Questions

**Comments to the Author**

1. If the authors have adequately addressed your comments raised in a previous round of review and you feel that this manuscript is now acceptable for publication, you may indicate that here to bypass the “Comments to the Author” section, enter your conflict of interest statement in the “Confidential to Editor” section, and submit your "Accept" recommendation.

Reviewer #1: (No Response)

Reviewer #2: (No Response)

Reviewer #3: (No Response)

2. Is the manuscript technically sound, and do the data support the conclusions?

Reviewer #1: Yes

Reviewer #2: Yes

Reviewer #3: Yes

3. Has the statistical analysis been performed appropriately and rigorously? 

Reviewer #1: Yes

Reviewer #2: Yes

Reviewer #3: I Don't Know

4. Have the authors made all data underlying the findings in their manuscript fully available?

Reviewer #1: Yes

Reviewer #2: Yes

Reviewer #3: Yes

5. Is the manuscript presented in an intelligible fashion and written in standard English?

Reviewer #1: No

Reviewer #2: No

Reviewer #3: Yes

6. Review Comments to the Author

Reviewer #1: The authors have put forth a great effort to address the comments of the reviewers, and should be commended. In particular, I appreciate the addition of the supplement as well as the clear outlining of quarantine strategies and model scenarios. I have some remaining minor comments, regarding clarity:

Introduction:

Substantial edits were made to the introduction in order to make the work more accessible to the general readership. However, some improvements could still be made to improve readability. In particular:

- Line 12 "Some extensions of SEIR-like models that account for spatial variability employed diffusion processes for the different sub-populations". It is unclear to me here what specifically is diffusing. I also don't think the reference to wildlife is particularly helpful. One suggestion-- these sentences might be replaced with sentences such as: "Some extensions of SEIR-like models account for spatial variability by employing diffusion processes on contact networks, in which an infection that appears at one node of a network spreads throughout the system through inter-node transmissions. However, this diffusive movement of disease through a population fails to capture human disease dynamics as human movement is not uniform".

- Line 21: is it possible to replace “such” with “homogenous”? I think it would be helpful to have this term introduced early on since it is highlighted throughout.

Results/Abstract:

- I appreciate the additional comparison of the infection heat maps to data from South Carolina. However, while reference to South Carolina appears in the abstract, no mention of it is in the text (with the exception of its reference in lines 70, 435, 475, 478). When referenced, it would be helpful if it would be clear that the "publicly available resources" or "heat maps appearing online" is from South Carolina.

- Relatedly, the citation for [35] appears to be annotated. I also was unable to visit the link and received an error. I understand there are challenges with changing links and the authors should not be held to this. If, however, it is possible to screenshot some instances of the video heat map to include in the supplement, this could alleviate this issue.

Discussion:

- The authors now use "sub-populations" to refer to the compartments of the SEPIR model. However, in many places (e.g., paragraph starting at line 422; paragraph starting at line 468) “populations” are still used rather than “sub-populations”

- Line 457: “supporting” our model conclusions, rather than “confirming” our model conclusions

- Line 492: define “red cities” instead of using jargon (e.g. "cities labeled 'red' for the highest risk tier")

Reviewer #2: The authors have carefully and successfully responded to my initial comments, and I believe the manuscript is much stronger. I only have a few minor remaining comments:

Line 51: Can you reword the hypothesis? Specifically, what are you referring to with “spatial growth of infected geographical domains”? Can you use more general language to make this clearer?

Line 69: What sorts of publicly available resources are you referring to here? It would be good to be more specific about the data products you are emulating and how those data can be used. A few specific examples such as the South Carolina Health department data would be useful in the text.

Line 11: I like this added text to provide context for the model but I think it could still use more detail. For example, as it is written, it isn’t clear why diffusion processes “fail to describe human behavior”. Could you describe why this is inappropriate more fully?

I appreciate the added sensitivity analyses for parameter values. Could you add a description of the results to the main text? What do you mean by “qualitative similarities” in the results? It would be good to describe specifically how the results change with different specified parameter values.

I like the statement that “the main purpose of the paper is to present our novel model to allow researchers working in close contact with authorities to apply it in their own countries” and believe it should be placed at the end of the Introduction section to make it clear what the scope of the paper is.

Table 1 is very useful to guide readers. Could the authors relabel the second and third columns using words? E.g., “Population density, n” and “initial infections” or something similar.

The figures and legends are much clearer now, but I am still unsure why the displayed time steps are different for each figure? I understand there are space constraints but maybe the authors could justify the choice of displayed time steps. I would also make sure that “x” and “y” are clearly defined in all figure legends.

There remain some grammar and spelling mistakes in the text. It would be good to carefully review.

Reviewer #3: The authors have made a number of important improvements, and clarified some of the issues which were brought up. There are some remaining issues to be addressed, however.

There are still some problematic claims about the existence of spatially heterogeneous techniques (e.g., "Moreover, these models do not include spatially-dependent infection spreading parameters, which are required to model geographically local quarantine.")

1. Metapopulation models routinely allow spatially-dependent infection parameters.

https://www.sciencedirect.com/science/article/pii/S0022519396900429

Part of the objection here is to the strength of the language used describing the limitations of existing models (particularly since the authors don't really engage the stochastic modelling literature much). Using a rich spatially heterogeneous parameterization of a compartmental epidemic model is not terribly difficult to do from a forward-simulation perspective, but often becomes computationally impractical when attempts at fitting a model to actual data are made (especially where formal inference is desired). It is certainly true that the vast majority of work published recently on COVID makes problematic assumptions like homogeneous mixing, but that's largely not a theoretical problem, but a practical one. This distinction should be clearer.

2. In the previous review, I asked that the discussion address "application of this technique to areas of sparse spread/inflated zero counts" - I apologize if I was insufficiently clear.

In response, the authors discuss initial conditions, but the question was intended to be more about the application. In real epidemics, regions of low and varying population density often give rise to chaotic, rather than smooth, disease outcomes - virtually identical communities with the same level of ongoing spread observe vastly (on a proportional scale) different outcomes due to things like cluster outbreaks (long term care facilities, in the case of COVID) and super-spreader events. Unless I'm mistaken, the proposed deterministic technique could run into trouble when fit to actual data in such regions, particularly where the grid produces regions of sparse population. To be clear, I'm not suggesting that this limitation be addressed by the current work, but I think that it should probably be addressed in the discussion.

3. Myself and several reviewers noted the issue of long-distance transmission, and the authors added the language: "It should be noted that when normal life conditions are restored, our model has to be modified to include far-distance traveling; this can be readily done using long distance infections, which we defer to future work."

I don't think this sufficiently addresses the concern. Even in areas with strict lockdowns, people routinely travel greater distances than 1km, whether essential workers or performing essential tasks like grocery shopping. Coupled with the smooth nature of the deterministic framework, this seems like a very important limitation worthy of additional discussion.

4. When addressing the lack of uniform testing, the authors added the language: "Incomplete testing data is not going to severely hamper our predictions so long as it is uniform in space (e.g., only 10% of the COVID-19 positives are detected everywhere), and in general we may expect this to be so within a certain country where a uniform testing policy is adopted. Obviously, absolute predicted numbers will not be obtained without the use of an adjusting factor between tested and predicted numbers."

There are a couple of things which come to mind that may be worth addressing here as well:

a. For COVID specifically, testing was non-uniform over time essentially everywhere, due to the availability of tests. In many places spatial non-uniformity was/is observed as well.

b. It's not clear to me that the lack of complete testing is a non-issue even if it is uniform - it seems that the dynamics on which the model is based perform very differently for different infectious fractions. It may be the case that the proposed model is sufficiently flexible to be fit to reported cases despite this limitation, but I'm not seeing that explicitly demonstrated in the manuscript. If I've missed something this would be helpful to clarify in the discussion section.

5. I also agree with the other reviewers that "population" and "sub-population" are confusing terms to use for disease states, since they conflict with the common practice of introducing strata into metapopulation style models (for example, age and geographic strata).

7. PLOS authors have the option to publish the peer review history of their article (what does this mean?). If published, this will include your full peer review and any attached files.

Reviewer #1: No

Reviewer #2: No

Reviewer #3: No

---

## [Author Response · Author response to Decision Letter 1]

12 Jan 2021

See attached Word file 'response_to_reviewers_2.docx'

---

## [Editor Report · Decision Letter 2]

13 Jan 2021

Epidemiological model for the inhomogeneous spatial spreading of COVID-19 and other diseases

PONE-D-20-21655R2

Dear Dr. Tsori,

We’re pleased to inform you that your manuscript has been judged scientifically suitable for publication and will be formally accepted for publication once it meets all outstanding technical requirements.

Kind regards,

Emanuele Giorgi

Academic Editor

PLOS ONE
---

## [Editor Report · Acceptance letter]

1 Feb 2021

PONE-D-20-21655R2 

Epidemiological model for the inhomogeneous spatial spreading of COVID-19 and other diseases  

Dear Dr. Tsori:

I'm pleased to inform you that your manuscript has been deemed suitable for publication in PLOS ONE. Congratulations! Your manuscript is now with our production department. 

Kind regards, 

on behalf of

Dr. Emanuele Giorgi 

Academic Editor

PLOS ONE